# CONTRASTIVE DIFFERENCE PREDICTIVE CODING

**Chongyi Zheng**
Carnegie Mellon University
chongyiz@andrew.cmu.edu

**Ruslan Salakhutdinov**
Carnegie Mellon University

**Benjamin Eysenbach**
Princeton University

## ABSTRACT

Predicting and reasoning about the future lie at the heart of many time-series questions. For example, goal-conditioned reinforcement learning can be viewed as learning representations to predict which states are likely to be visited in the future. While prior methods have used contrastive predictive coding to model time series data, learning representations that encode long-term dependencies usually requires large amounts of data. In this paper, we introduce a temporal difference version of contrastive predictive coding that stitches together pieces of different time series data to decrease the amount of data required to learn predictions of future events. We apply this representation learning method to derive an off-policy algorithm for goal-conditioned RL. Experiments demonstrate that, compared with prior RL methods, ours achieves $2\times$ median improvement in success rates and can better cope with stochastic environments. In tabular settings, we show that our method is about $20\times$ more sample efficient than the successor representation and $1500\times$ more sample efficient than the standard (Monte Carlo) version of contrastive predictive coding.

**Code**: https://github.com/chongyi-zheng/td_infonce
**Website**: https://chongyi-zheng.github.io/td_infonce

## 1 INTRODUCTION

Learning representations is important for modeling high-dimensional time series data. Many applications of time-series modeling require representations that not only contain information about the contents of a particular observation, but also about how one observation relates to others that co-occur in time. Acquiring representations that encode temporal information is challenging, especially when attempting to capture long-term temporal dynamics: the frequency of long-term events may decrease with the time scale, meaning that learning longer-horizon dependencies requires larger quantities of data.

In this paper, we study contrastive representation learning on time series data – positive examples co-occur nearby in time, so the distances between learned representations should encode the likelihood of transiting from one representation to another. Building on prior

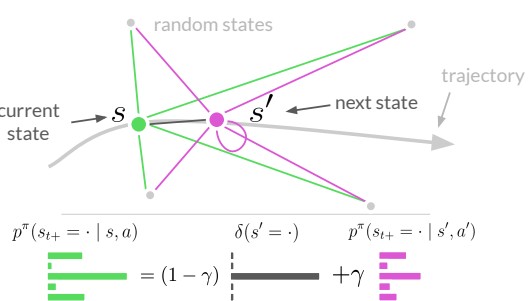

Figure 1: **TD InfoNCE** is a nonparametric version of the successor representation. *(Top)* The distances between learned representations indicate the probability of transitioning to the next state and a set of randomly-sampled states. *(Bottom)* We update these representations so they assign high likelihood to *(a)* the next state and *(b)* states likely to be visited after the next state. See Sec. 3 for details.

work that uses the InfoNCE (Sohn, 2016; Oord et al., 2018) loss to learn representations of time-series data effectively, we will aim to build a temporal difference version of this loss. Doing so allows us to optimize this objective with fewer samples, enables us to stitch together pieces of different time series data, and enables us to perform counterfactual reasoning – we should be able to estimate which representations we would have learned, if we had collected data in a different way.

After a careful derivation, our resulting method can be interpreted as a non-parametric form of the successor representation (Dayan, 1993), as shown in Fig. 1.

The main contribution of this paper is a temporal difference estimator for InfoNCE. We then apply this estimator to develop a new algorithm for goal-conditioned RL. Experiments on both state-based and image-based benchmarks show that our algorithm outperforms prior methods, especially on the most challenging tasks. Additional experiments demonstrate that our method can handle stochasticity in the environment more effectively than prior methods. We also demonstrate that our algorithm can be effectively applied in the offline setting. Additional tabular experiments demonstrate that TD InfoNCE is up to $1500\times$ more sample efficient than the standard Monte Carlo version of the loss and that it can effectively stitch together pieces of data.

## 2 RELATED WORK

This paper will study the problem of self-supervised RL, building upon prior methods on goal-condition RL, contrastive representation learning, and methods for predicting future state visitations. Our analysis will draw a connection between these prior methods, a connection which will ultimately result in a new algorithm for goal-conditioned RL. We discuss connections with unsupervised skill learning and mutual information in Appendix B.

**Goal-conditioned reinforcement learning.** Prior work has proposed many frameworks for learning goal-conditioned policies, including conditional supervised learning (Ding et al., 2019; Ghosh et al., 2020; Gupta et al., 2020; Emmons et al., 2021; Lynch et al., 2020; Oh et al., 2018; Sun et al., 2019), actor-critic methods Andrychowicz et al. (2017); Nachum et al. (2018); Chane-Sane et al. (2021), semi-parametric planning (Pertsch et al., 2020; Fang et al., 2022; 2023; Eysenbach et al., 2019; Nair & Finn, 2019; Gupta et al., 2020), and distance metric learning (Wang et al., 2023; Tian et al., 2020a; Nair et al., 2020b; Durugkar et al., 2021). These methods have demonstrated impressive results on a range of tasks, including real-world robotic tasks (Ma et al., 2022; Shah et al., 2022; Zheng et al., 2023). While some methods require manually-specified reward functions or distance functions, our work builds upon a self-supervised interpretation of goal-conditioned RL that casts this problem as predicting which states are likely to be visited in the future (Eysenbach et al., 2020; 2022; Blier et al., 2021).

**Contrastive representation learning.** Contrastive learning methods have become a key tool for learning representations in computer vision and NLP (Chopra et al., 2005; Schroff et al., 2015; Sohn, 2016; Oh Song et al., 2016; Wang & Isola, 2020; Oord et al., 2018; Tschannen et al., 2019; Weinberger & Saul, 2009; He et al., 2022; Radford et al., 2021; Chen et al., 2020; Tian et al., 2020b; Gao et al., 2021). These methods assign similar representations to positive examples and dissimilar representations to negative examples or outdated embeddings (Grill et al., 2020). The two main contrastive losses are based on binary classification ("NCE") ranking loss ("InfoNCE") (Ma & Collins, 2018). Modern contrastive learning methods typically employ the ranking-based objective to learn representations of images (Chen et al., 2020; Tian et al., 2020b; Henaff, 2020; Wu et al., 2018), text (Logeswaran & Lee, 2018; Jia et al., 2021; Radford et al., 2021) and sequential data (Nair et al., 2022; Sermanet et al., 2018). Prior works have also provided theoretical analysis for these methods from the perspective of mutual information maximization (Linsker, 1988; Poole et al., 2019), noise contrastive estimation (Gutmann & Hyvärinen, 2010; Ma & Collins, 2018; Tsai et al., 2020; Arora et al., 2019), and the geometry of the learned representations (Wang & Isola, 2020). In the realm of RL, prior works have demonstrated that contrastive methods can provide effective reward functions and auxiliary learning objectives (Laskin et al., 2020a;b; Hansen-Estruch et al., 2022; Choi et al., 2021; Nair et al., 2020a; 2018), and can also be used to formulate the goal-reaching problem in an entirely self-supervised manner (Ma et al., 2022; Durugkar et al., 2021; Eysenbach et al., 2020; 2022). Our method will extend these results by building a temporal difference version of the "ranking"-based contrastive loss; this loss will enable us to use data from one policy to estimate which states a different policy will visit.

**Temporal difference learning and successor representation.** Another line of work studies using temporal difference learning to predict states visited in the future, building upon successor representations and successor features (Dayan, 1993; Barreto et al., 2017; 2019; Blier et al., 2021).

While learning successor representation using temporal difference bears a similarity to the typical Q-Learning algorithm (Watkins & Dayan, 1992; Fu et al., 2019; Mnih et al., 2015) in the tabular setting, directly estimating this quantity is difficult with continuous states and actions (Janner et al., 2020; Barreto et al., 2017; Touati & Ollivier, 2021; Blier et al., 2021). To lift this limitation, we will follow prior work (Eysenbach et al., 2022; 2020; Touati & Ollivier, 2021) in predicting the successor representation indirectly: rather than learning a representation whose coordinates correspond to visitation probabilities, we will learn state representations such that their inner product corresponds to a visitation probability. Unlike prior methods, we will show how the common InfoNCE objective can be estimated in a temporal difference fashion, opening the door to off-policy reasoning and enabling our method to reuse historical data to improve data efficiency.

## 3 METHOD

We start by introducing notation and prior approaches to the contrastive representation learning and the goal-conditioned RL problems. We then propose a new self-supervised actor-critic algorithm that we will use in our analysis.

### 3.1 PRELIMINARIES

We first review prior work in contrastive representation learning and goal-conditioned RL. Our method will use ideas from both.

**Contrastive representation via InfoNCE.** Contrastive representation learning aims to learn a representation space, pushing representations of positive examples together and pushing representations of negative examples away. InfoNCE (also known as contrastive predictive coding) (Sohn, 2016; Jozefowicz et al., 2016; Oord et al., 2018; Henaff, 2020) is a widely used contrastive loss, which builds upon noise contrastive estimation (NCE) (Gutmann & Hyvärinen, 2010; Ma & Collins, 2018). Given the distribution of data $p_\mathcal{X}(x), p_\mathcal{Y}(y)$ over data $x \in \mathcal{X}, y \in \mathcal{Y}$ and the conditional distribution of positive pairs $p_{\mathcal{Y}|\mathcal{X}}(y|x)$ over $\mathcal{X} \times \mathcal{Y}$, we sample $x \sim p_\mathcal{X}(x)$, $y^{(1)} \sim p_{\mathcal{Y}|\mathcal{X}}(y \mid x)$, and $\{y^{(2)}, \cdots, y^{(N)}\} \sim p_\mathcal{Y}(y)$. The InfoNCE loss is defined as

$$\mathcal{L}_{\text{InfoNCE}}(f) \triangleq \mathbb{E}_{\substack{x \sim p_\mathcal{X}(x), y^{(1)} \sim p_{\mathcal{Y}|\mathcal{X}}(y|x) \\ y^{(2:N)} \sim p_\mathcal{Y}(y)}} \left[ \log \frac{e^{f(x,y^{(1)})}}{\sum_{i=1}^N e^{f(x,y^{(i)})}} \right], \tag{1}$$

where $f : \mathcal{X} \times \mathcal{Y} \mapsto \mathbb{R}$ is a parametric function. Following prior work (Eysenbach et al., 2022; Wang & Isola, 2020; Touati & Ollivier, 2021), we choose to parameterize $f(\cdot, \cdot)$ via the inner product of representations of data $f(x, y) = \phi(x)^\top \psi(y)$, where $\phi(\cdot)$ and $\psi(\cdot)$ map data to $\ell_2$ normalized vectors of dimension $d$. We will call $f$ the *critic function* and $\phi$ and $\psi$ the *contrastive representations*. The Bayes-optimal critic for the InfoNCE loss satisfies (Poole et al., 2019; Ma & Collins, 2018; Oord et al., 2018)

$$\exp\left(f^\star(x, y)\right) = \frac{p(y \mid x)}{p(y)c(x)},$$

where $c(\cdot)$ is an arbitrary function. We can estimate this arbitrary function using the optimal critic $f^\star$ by sampling multiple negative pairs from the data distribution:

$$\mathbb{E}_{p(y)}\left[\exp\left(f^\star(x, y)\right)\right] = \int \cancel{p(y)} \frac{p(y \mid x)}{\cancel{p(y)}c(x)} dy = \frac{1}{c(x)} \underbrace{\int p(y \mid x) dy}_{=1} = \frac{1}{c(x)}. \tag{2}$$

**Reinforcement learning and goal-conditioned RL.** We will consider a Markov decision process defined by states $s \in \mathcal{S}$, actions $a \in \mathcal{A}$, rewards $r : \mathcal{S} \times \mathcal{A} \times \mathcal{S} \mapsto \mathbb{R}$. Using $\Delta(\cdot)$ denotes the probability simplex, we define an initial state distribution $p_0 : \mathcal{S} \mapsto \Delta(\mathcal{S})$, discount factor $\gamma \in (0, 1]$, and dynamics $p : \mathcal{S} \times \mathcal{A} \mapsto \Delta(\mathcal{S})$. Given a policy $\pi : \mathcal{S} \mapsto \Delta(\mathcal{A})$, we will use $p_t^\pi(s_{t+} \mid s, a)$ to denote the probability density of reaching state $s_{t+}$ after exactly $t$ steps, starting at state $s$ and action $a$ and then following the policy $\pi(a \mid s)$. We can then define the discounted state

occupancy measure (Ho & Ermon, 2016; Zhang et al., 2020; Eysenbach et al., 2020; 2022; Zheng et al., 2023) starting from state $s$ and action $a$ as

$$p^\pi(s_{t+} \mid s, a) \triangleq (1 - \gamma) \sum_{t=1}^{\infty} \gamma^{t-1} p_t^\pi(s_{t+} \mid s, a). \tag{3}$$

Prior work (Dayan, 1993) have shown that this discounted state occupancy measure follows a recursive relationship between the density at the current time step and the future time steps:

$$p^\pi(s_{t+} \mid s, a) = (1 - \gamma)p(s' = s_{t+} \mid s, a) + \gamma \mathbb{E}_{\substack{s' \sim p(s'\mid s,a) \\ a' \sim \pi(a'\mid s')}} \left[ p^\pi(s_{t+} \mid s', a') \right]. \tag{4}$$

For goal-conditioned RL, we define goals $g \in \mathcal{S}$ in the same space as states and consider a goal-conditioned policy $\pi(a \mid s, g)$ and the corresponding goal-conditioned discounted state occupancy measure $p^\pi(s_{t+} \mid s, a, g)$. For evaluation, we will sample goals from a distribution $p_g : \mathcal{S} \mapsto \Delta(\mathcal{S})$. Following prior work (Eysenbach et al., 2020; Rudner et al., 2021), we define the objective of the goal-reaching policy as maximizing the probability of reaching desired goals under its discounted state occupancy measure while commanding the same goals:

$$\max_{\pi(\cdot\mid\cdot,\cdot)} \mathbb{E}_{p_g(g), p_0(s), \pi(a\mid s,g)} \left[ p^\pi(s_{t+} = g \mid s, a, g) \right]. \tag{5}$$

In tabular settings, this objective is the same as maximizing expected returns using a sparse reward function $r(s, a, s', g) = (1 - \gamma)\delta(s' = g)$ (Eysenbach et al., 2022). Below, we review two strategies for estimating the discounted state occupancy measure. Our proposed method (Sec. 3.2) will combine the strengths of these methods while lifting their respective limitations.

**Contrastive RL and C-Learning.**    Our focus will be on using contrastive representation learning to build a new goal-conditioned RL algorithm, following a template set in prior work (Eysenbach et al., 2022; 2020). These *contrastive RL* methods are closely related to the successor representation (Dayan, 1993): they aim to learn representations whose inner products correspond to the likelihoods of reaching future states. Like the successor representation, representations from these contrastive RL methods can then be used to represent the Q function for any reward function (Mazoure et al., 2022). Prior work (Eysenbach et al., 2022) has shown how both NCE and the InfoNCE losses can be used to derive Monte Carlo algorithms for estimating the discounted state occupancy measure. We review the Monte Carlo InfoNCE loss below. Given a policy $\pi(a \mid s)$, consider learning contrastive representations for a state and action pair $x = (s, a)$ and a potential future state $y = s_{t+}$. We define the data distribution to be the joint distribution of state-action pairs $p_{\mathcal{X}}(x) = p(s, a)$ and the marginal distribution of future states $p_{\mathcal{Y}}(y) = p(s_{t+})$, representing either the distribution of a replay buffer (online) or the distribution of a dataset (offline). The conditional distribution of positive pairs is set to the discounted state occupancy measure for policy $\pi$, $p_{\mathcal{Y}\mid\mathcal{X}}(y \mid x) = p^\pi(s_{t+} \mid s, a)$, resulting in a Monte Carlo (MC) estimator

$$\mathcal{L}_{\text{MC InfoNCE}}(f) = \mathbb{E}_{\substack{(s,a) \sim p(s,a), s_{t+}^{(1)} \sim p^\pi(s_{t+}\mid s,a) \\ s_{t+}^{(2:N)} \sim p(s_{t+})}} \left[ \log \frac{e^{f(s,a,s_{t+}^{(1)})}}{\sum_{i=1}^{N} e^{f(s,a,s_{t+}^{(i)})}} \right] \tag{6}$$

and an optimal critic function satisfying

$$\exp(f^\star(s, a, s_{t+})) = \frac{p^\pi(s_{t+} \mid s, a)}{p(s_{t+})c(s, a)}. \tag{7}$$

This loss estimates the discounted state occupancy measure in a Monte Carlo manner. Computing this estimator usually requires sampling future states from the discounted state occupancy measure of the policy $\pi$, i.e., on-policy data. While, in theory, Monte Carlo estimator can be used in an off-policy manner by applying importance weights to correct actions, this estimator usually suffers from high variance and is potentially sample inefficient than temporal difference methods (Precup et al., 2000; 2001).

In the same way that temporal difference (TD) algorithms tend to be more sample efficient than Monte Carlo algorithms for reward maximization (Sutton & Barto, 2018), we expect that TD contrastive methods are more sample efficient at estimating probability ratios than their Monte Carlo counterparts. Given that the InfoNCE tends to outperform the NCE objective in other machine learning disciplines, we conjecture that our TD InfoNCE objective will outperform the TD NCE objective (Eysenbach et al., 2020) (see experiments in Appendix. E.3).

## 3.2 Temporal Difference InfoNCE

In this section, we derive a new loss for estimating the discounted state occupancy measure for a fixed policy. This loss will be a temporal difference variant of the InfoNCE loss. We will use **temporal difference InfoNCE (TD InfoNCE)** to refer to our loss function.

In the off-policy setting, we aim to estimate the discounted state occupancy measure of the policy $\pi$ given a dataset of transitions $\mathcal{D} = \{(s, a, s')_i\}_{i=1}^{D}$ collected by another behavioral policy $\beta(a \mid s)$. This setting is challenging because we do not obtain samples from the discounted state occupancy measure of the target policy $\pi$. Addressing this challenge involves two steps: *(i)* expanding the MC estimator (Eq. 6) via the recursive relationship of the discounted state occupancy measure (Eq. 4), and *(ii)* estimating the expectation over the discounted state occupancy measure via importance sampling. We first use the identity from Eq. 4 to express the MC InfoNCE loss as the sum of a next-state term and a future-state term:

$$
\mathbb{E}_{\substack{(s,a)\sim p(s,a) \\ s_{t+}^{(2:N)}\sim p(s_{t+})}} \left[ (1-\gamma) \underbrace{\mathbb{E}_{s_{t+}^{(1)}\sim p(s'|s,a)} \left[ \log \frac{e^{f(s,a,s_{t+}^{(1)})}}{\sum_{i=1}^{N} e^{f(s,a,s_{t+}^{(i)})}} \right]}_{\mathcal{L}_1(f)} \right.
$$
$$
\left. + \gamma \underbrace{\mathbb{E}_{\substack{s'\sim p(s'|s,a),a'\sim\pi(a'|s') \\ s_{t+}^{(1)}\sim p^{\pi}(s_{t+}|s',a')}} \left[ \log \frac{e^{f(s,a,s_{t+}^{(1)})}}{\sum_{i=1}^{N} e^{f(s,a,s_{t+}^{(i)})}} \right]}_{\mathcal{L}_2(f)} \right].
$$

While this estimate is similar to a TD target for Q-Learning (Watkins & Dayan, 1992; Fu et al., 2019), the second term requires sampling from the discounted state occupancy measure of policy $\pi$. To avoid this sampling, we next replace the expectation over $p^{\pi}(s_{t+} \mid s', a')$ in $\mathcal{L}_2(f)$ by an importance weight,

$$
\mathcal{L}_2(f) = \mathbb{E}_{\substack{s'\sim p(s'|s,a),a'\sim\pi(a'|s') \\ s_{t+}^{(1)}\sim p(s_{t+})}} \left[ \frac{p^{\pi}(s_{t+}^{(1)} \mid s', a')}{p(s_{t+}^{(1)})} \log \frac{e^{f(s,a,s_{t+}^{(1)})}}{\sum_{i=1}^{N} e^{f(s,a,s_{t+}^{(i)})}} \right].
$$

If we could estimate the importance weight, then we could easily estimate this term by sampling from $p(s_{t+})$. We will estimate this importance weight by rearranging the expression for the optimal critic (Eq. 7) and substituting our estimate for the normalizing constant $c(s, a)$ (Eq. 2):

$$
\frac{p^{\pi}(s_{t+}^{(1)} \mid s, a)}{p(s_{t+}^{(1)})} = c(s, a) \cdot \exp\left( f^{\star}(s, a, s_{t+}^{(1)}) \right) = \frac{e^{f^{\star}(s,a,s_{t+}^{(1)})}}{\mathbb{E}_{p(s_{t+})}\left[ e^{f^{\star}(s,a,s_{t+})} \right]}. \tag{8}
$$

We will use $w(s, a, s_{t+}^{(1:N)})$ to denote our estimate of this, using $f$ in place of $f^{\star}$ and using a finite-sample estimate of the expectation in the denominator:

$$
w(s, a, s_{t+}^{(1:N)}) \triangleq \frac{e^{f(s,a,s_{t+}^{(1)})}}{\frac{1}{N}\sum_{i=1}^{N} e^{f(s,a,s_{t+}^{(i)})}} \tag{9}
$$

This weight accounts for the effect of the discounted state occupancy measure of the target policy. Additionally, it corresponds to the categorical classifier that InfoNCE produces (without constant $N$). Taken together, we can now substitute the importance weight in $\mathcal{L}_2(f)$ with our estimate in Eq. 9, yielding a temporal difference (TD) InfoNCE estimator

$$
\mathcal{L}_{\text{TD InfoNCE}}(f) \triangleq \mathbb{E}_{\substack{(s,a)\sim p(s,a) \\ s_{t+}^{(2:N)}\sim p(s_{t+})}} \left[ (1-\gamma)\mathbb{E}_{s_{t+}^{(1)}\sim p(s'|s,a)} \left[ \log \frac{e^{f(s,a,s_{t+}^{(1)})}}{\sum_{i=1}^{N} e^{f(s,a,s_{t+}^{(i)})}} \right] \right.
$$
$$
\left. +\gamma\mathbb{E}_{\substack{s'\sim p(s'|s,a) \\ a'\sim\pi(a'|s') \\ s_{t+}^{(1)}\sim p(s_{t+})}} \left[ \lfloor w(s', a', s_{t+}^{(1:N)}) \rfloor_{\text{sg}} \log \frac{e^{f(s,a,s_{t+}^{(1)})}}{\sum_{i=1}^{N} e^{f(s,a,s_{t+}^{(i)})}} \right] \right], \tag{10}
$$

---

**Algorithm 1** Temporal Difference InfoNCE. We use $\mathcal{CE}$ to denote the cross entropy loss, taken across the rows of a matrix of logits and labels. We use $F$ as a matrix of logits, where $F[i, j] = \phi(s_t^{(i)}, a_t^{(i)}, g^{(i)})^\top \psi(s_{t+}^{(j)})$. See Appendix D.1 for details.

---

1: **Input** contrastive representations $\phi_\theta$ and $\psi_\theta$, target representations $\phi_{\bar\theta}$ and $\psi_{\bar\theta}$, and goal-conditioned policy $\pi_\omega$.
2: **for** each iteration **do**
3:     Sample $\{(s_t^{(i)}, a_t^{(i)}, s_{t+1}^{(i)}, g^{(i)}, s_{t+}^{(i)})\}_{i=1}^N \sim$ replay buffer / dataset, $a^{(i)} \sim \pi(a \mid s_t^{(i)}, g^{(i)})$.
4:     Compute $F_{\text{next}}, F_{\text{future}}, F_{\text{goal}}$ using $\phi_\theta$ and $\psi_\theta$.
5:     Compute $\bar F_w$ using $\phi_{\bar\theta}$ and $\psi_{\bar\theta}$.
6:     $W \leftarrow N \cdot \texttt{stop\_grad}\big(\textsc{Softmax}(\bar F_w)\big)$
7:     $\mathcal{L}(\theta) \leftarrow (1 - \gamma)\mathcal{CE}(\text{logits} = F_{\text{next}}, \text{labels} = I_N) + \gamma\mathcal{CE}(\text{logits} = F_{\text{future}}, \text{labels} = W)$
8:     $\mathcal{L}(\omega) \leftarrow \mathcal{CE}(\text{logits} = F_{\text{goal}}, \text{labels} = I_N)$
9:     Update $\theta, \omega$ by taking gradients of $\mathcal{L}(\theta), \mathcal{L}(\omega)$.
10:     Update $\bar\theta$ using an exponential moving average.
11: **Return** $\phi_\theta, \psi_\theta$, and $\pi_\omega$.

---

where $\lfloor \cdot \rfloor_{\text{sg}}$ indicates the gradient of the importance weight should not affect the gradient of the entire objective. As shown in Fig. 1, we can interpret the first term as pulling together the representations of the current state-action pair $\phi(s, a)$ and the next state $\psi(s')$; the second term pulls the representations at the current step $\phi(s, a)$ similar to the (weighted) predictions from the future state $\psi(s_{t+})$. Importantly, the TD InfoNCE estimator is equivalent to the MC InfoNCE estimator for the optimal critic function: $\mathcal{L}_{\text{TD InfoNCE}}(f^\star) = \mathcal{L}_{\text{MC InfoNCE}}(f^\star)$.

**Convergence and connections.** In Appendix A, we prove that optimizing a variant of the TD InfoNCE objective is equivalent to perform one step policy evaluation with a new Bellman operator; thus, repeatedly optimizing this objective yields the correct discounted state occupancy measure. This analysis considers the tabular setting and assumes that the denominators of the softmax functions and $w$ in Eq. 10 are computed using an exact expectation. We discuss the differences between TD InfoNCE and C-learning (Eysenbach et al., 2020) (a temporal difference estimator of the NCE objective) in Appendix E.3. Appendix C discusses how TD InfoNCE corresponds to a nonparametric variant of the successor representation.

### 3.3 GOAL-CONDITIONED POLICY LEARNING

The TD InfoNCE method provides a way for estimating the discounted state occupancy measure. This section shows how this estimator can be used to derive a new algorithm for goal-conditioned RL. This algorithm will alternate between *(1)* estimating the occupancy measure using the TD InfoNCE objective and *(2)* optimizing the policy to maximize the likelihood of the desired goal under the estimated occupancy measure. Pseudo-code is shown in Algorithm 1, and additional details are in Appendix D.1, and code is available online.[1]

While our TD InfoNCE loss in Sec. 3.2 estimates the discounted state occupancy measure for policy $\pi(a \mid s)$, we can extend it to the goal-conditioned setting by replacing $\pi(a \mid s)$ with $\pi(a \mid s, g)$ and $f(s, a, s_{t+})$ with $f(s, a, g, s_{t+})$, resulting in a goal-conditioned TD InfoNCE estimator. This goal-conditioned TD InfoNCE objective estimates the discounted state occupancy measure of *any* future state for a goal-conditioned policy commanding *any* goal. Recalling that the discounted state occupancy measure corresponds to the Q function (Eysenbach et al., 2022), the policy objective is to select actions that maximize the likelihood of the commanded goal:

$$\mathbb{E}_{\substack{p_g(g), p_0(s) \\ \pi(a_0 \mid s, g)}} \left[\log p^\pi(s_{t+} = g \mid s, a, g)\right] = \mathbb{E}_{\substack{g \sim p_g(g), s \sim p_0(s) \\ a_0 \sim \pi(a \mid s, g), s_{t+}^{(1:N)} \sim p(s_{t+})}} \left[\log \frac{e^{f^\star(s, a, g, s_{t+} = g)}}{\sum_{i=1}^N e^{f^\star(s, a, g, s_{t+}^{(i)})}}\right]. \tag{11}$$

In practice, we optimize both the critic function and the policy for one gradient step iteratively, using our estimated $f$ in place of $f^\star$.

---

[1]https://github.com/chongyi-zheng/td_infonce

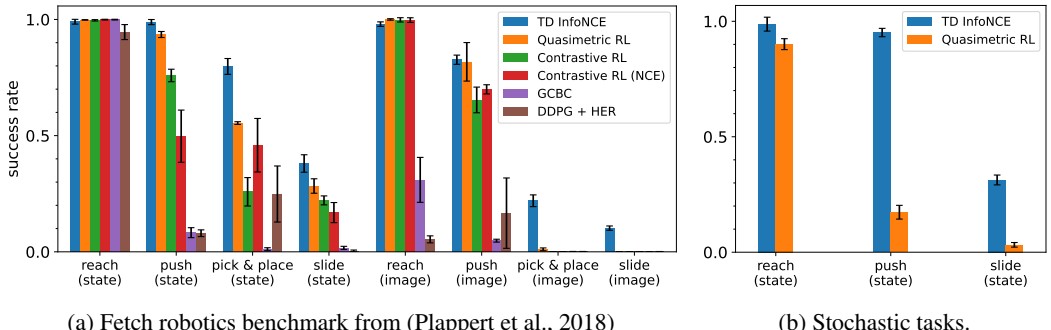

(a) Fetch robotics benchmark from (Plappert et al., 2018)    (b) Stochastic tasks.

Figure 2: **Evaluation on online GCRL benchmarks.** *(Left)* TD InfoNCE performs similarly to or outperforms all baselines on both state-based and image-based tasks. *(Right)* On stochastic versions of the state-based tasks, TD InfoNCE outperforms the strongest baseline (QRL). Appendix Fig. 6 shows the learning curves.

## 4  EXPERIMENTS

Our experiments start with comparing goal-conditioned TD InfoNCE to prior goal-conditioned RL approaches on both online and offline goal-conditioned RL (GCRL) benchmarks. We then analyze the properties of the critic function and the policy learned by this method. Visualizing the representations learned by TD InfoNCE reveals that linear interpolation corresponds to a form of planning. Appendix E.3 ablates the difference between TD InfoNCE and a prior temporal difference method based on NCE. All experiments show means and standard deviations over five random seeds.

### 4.1  COMPARING TO PRIOR GOAL-CONDITIONED RL METHODS

We compare TD InfoNCE to four baselines on an online GCRL benchmark (Plappert et al., 2018) containing four manipulation tasks for the Fetch robot. The observations and goals of those tasks can be either a state of the robot and objects or a $64 \times 64$ RGB image. We will evaluate using both versions. The first baseline, Quasimetric Reinforcement Learning (QRL) (Wang et al., 2023), is a state-of-the-art approach that uses quasimetric models to learn the optimal goal-conditioned value functions and the corresponding policies. The second baseline is contrastive RL (Eysenbach et al., 2022), which estimates the discounted state occupancy measure using $\mathcal{L}_{\text{MC InfoNCE}}$ (Eq. 6). Our third baseline is a variant of contrastive RL (Eysenbach et al., 2022) using binary NCE loss. We call this method contrastive RL (NCE). The fourth baseline is the goal-conditioned behavioral cloning (GCBC) (Ding et al., 2019; Emmons et al., 2021; Ghosh et al., 2020; Lynch et al., 2020; Sun et al., 2019; Srivastava et al., 2019). We also include a comparison with an off-the-shelf actor-critic algorithm augmented with hindsight relabeling (Andrychowicz et al., 2017; Levy et al., 2018; Riedmiller et al., 2018; Schaul et al., 2015) to learn a goal-conditioned policy (DDPG + HER).

We report results in Fig. 2a, and defer the full learning curves to Appendix Fig. 6. These results show that TD InfoNCE matches or outperforms other baselines on all tasks, both for state and image observations. On those more challenging tasks (`pick & place (state / image)` and `slide (state / image)`), TD InfoNCE achieves a $2\times$ median improvement relative to the strongest baseline (Appendix Fig. 6). On the most challenging tasks, image-based `pick & place` and `slide`, TD InfoNCE is the only method achieving non-negligible success rates. For those tasks where the success rate fails to separate different methods significantly (e.g., `slide (state)` and `push (image)`), we include comparisons using minimum distances of the gripper or the object to the goal over an episode in Appendix Fig. 7. We speculate this observation is because TD InfoNCE estimates the discounted state occupancy measure more accurately, a hypothesis we will investigate in Sec. 4.3.

Among those baselines, QRL is the strongest one. Unlike TD InfoNCE, the derivation of QRL assumes the dynamics are deterministic. This difference motivates us to study whether TD InfoNCE continues achieving high success rates in environments with stochastic noise. To study this, we compare TD InfoNCE to QRL on a variant of the Fetch benchmark where observations are corrupted with probability $0.1$. As shown in Fig. 2b, TD InfoNCE maintains high success rates while the performance of QRL decreases significantly, suggesting that TD InfoNCE can better cope with stochasticity in the environment.

Table 1: Evaluation on offline D4RL AntMaze benchmarks.

| | TD InfoNCE | QRL | Contrastive RL | GCBC | DT | IQL | TD3 + BC |
|---|---|---|---|---|---|---|---|
| umaze-v2 | $84.9 \pm 1.2$ | $76.8 \pm 2.3$ | $79.8 \pm 1.6$ | 65.4 | 65.6 | **87.5** | 78.6 |
| umaze-diverse-v2 | $\mathbf{91.7 \pm 1.3}$ | $80.1 \pm 1.3$ | $77.6 \pm 2.8$ | 60.9 | 51.2 | 62.2 | 71.4 |
| medium-play-v2 | $\mathbf{86.8 \pm 1.7}$ | $76.5 \pm 2.1$ | $72.6 \pm 2.9$ | 58.1 | 1.0 | 71.2 | 10.6 |
| medium-diverse-v2 | $\mathbf{82.0 \pm 3.4}$ | $73.4 \pm 1.9$ | $71.5 \pm 1.3$ | 67.3 | 0.6 | 70.0 | 3.0 |
| large-play-v2 | $47.0 \pm 2.5$ | $\mathbf{52.9 \pm 2.8}$ | $48.6 \pm 4.4$ | 32.4 | 0.0 | 39.6 | 0.2 |
| large-diverse-v2 | $55.6 \pm 3.6$ | $51.5 \pm 3.8$ | $\mathbf{54.1 \pm 5.5}$ | 36.9 | 0.2 | 47.5 | 0.0 |

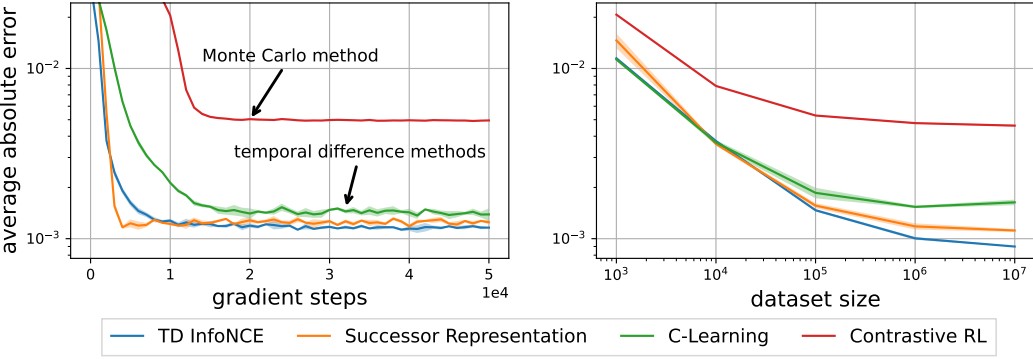

Figure 3: **Estimating the discounted state occupancy measure in a tabular setting.** *(Left)* Temporal difference methods have lower errors than the Monte Carlo method. Also note that our TD InfoNCE converges as fast as the best baseline (successor representation). *(Right)* TD InfoNCE is more data efficient than other methods. Using a dataset of size 10M, TD InfoNCE achieves an error rate 25% lower than the best baseline; TD InfoNCE also matches the performance of C-learning with 130× less data.

## 4.2 EVALUATION ON OFFLINE GOAL REACHING

We next study whether the good performance of TD InfoNCE transfers to the setting without any interaction with the environment (i.e., offline RL). We evaluate on AntMaze tasks from the D4RL benchmark (Fu et al., 2020). The results in Table 1 show that TD InfoNCE outperforms most baselines on most tasks. See Appendix D.3 for details.

## 4.3 ACCURACY OF THE ESTIMATED DISCOUNTED STATE OCCUPANCY MEASURE

This section tests the hypothesis that our TD InfoNCE loss will be more accurate and sample efficient than alternative Monte Carlo methods (namely, contrastive RL (Eysenbach et al., 2022)) in predicting the discounted state occupancy measure. We will use the tabular setting so that we can get a ground truth estimate. We compare TD InfoNCE to three baselines. Successor representations (Dayan, 1993) can also be learned in a TD manner, though can be challenging to apply beyond tabular settings. C-learning is similar to TD InfoNCE in that it uses a temporal difference method to optimize a contrastive loss, but differs in using a binary cross entropy loss instead of a softmax cross entropy loss. Contrastive RL is the MC counterpart of TD InfoNCE. We design a $5 \times 5$ gridworld with 125 states and 5 actions (up, down, left, right, and no-op) and collect 100K transitions using a uniform random policy, $\mu(a \mid s) = \text{UNIF}(\mathcal{A})$. We evaluate each method by measuring the absolute error between the predicted probability $\hat{p}$ and the ground truth probability $p^{\mu}$, averaging over all pairs of $(s, a, s_{t+})$:

$$\frac{1}{|\mathcal{S}||\mathcal{A}||\mathcal{S}|} \sum_{s,a,s_{t+}} |\hat{p}(s_{t+} \mid s, a) - p^{\mu}(s_{t+} \mid s, a)|.$$

For the three TD methods, we compute the TD target in a SARSA manner (Sutton & Barto, 2018). For those methods estimating a probability ratio, we convert the prediction to a probability by multiplying by the empirical state marginal. Results in Fig. 3 show that TD methods achieve lower errors than the Monte Carlo method, while TD InfoNCE converges faster than C-Learning. Appendix E.2 discusses why all methods plateau above zero.

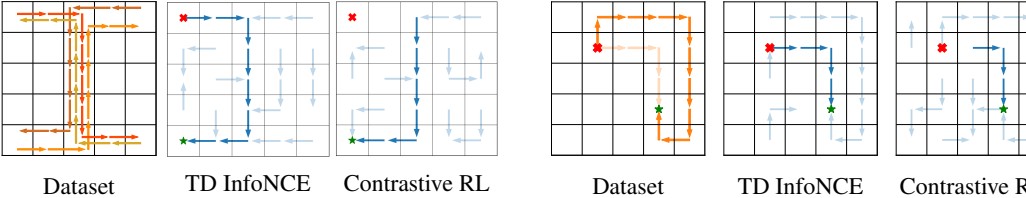

| Dataset | TD InfoNCE | Contrastive RL | | Dataset | TD InfoNCE | Contrastive RL |

Figure 4: **Stitching trajectories in a dataset.** The behavioral policy collects "Z" style trajectories. Unlike the Monte Carlo method (contrastive RL) , our TD InfoNCE successfully "stitches" these trajectories together, navigating between pairs of (start ✖, goal ★) states unseen in the training trajectories. Appendix Fig. 8 shows additional examples.

Figure 5: **Searching for shortcuts in skewed datasets.** *(Left)* Conditioned on different initial states ✖ and goals ★, we collect datasets with 95% long paths (dark) and 5% short paths (light). *(Center)* TD InfoNCE infers the shortest path, *(Right)* while contrastive RL fails to find this path. Appendix Fig. 9 shows additional examples.

Our next experiments studies sample efficiency. We hypothesize that the softmax in the TD InfoNCE loss may provide more learning signal than alternative methods, allowing it to achieve lower error on a fixed budget of data. To test this hypothesis, we run experiments with dataset sizes from 1K to 10M on the same gridworld, comparing TD InfoNCE to the same set of baselines. We report results in Fig. 3 with errors showing one standard deviation after training for 50K gradient steps for each approach. These results suggest that methods based on temporal difference learning predict more accurately than Monte Carlo method when provided with the same amount of data. Compared with its Monte Carlo counterpart, TD InfoNCE is $1500\times$ more sample efficient ($6.5 \times 10^3$ vs $10^7$ transitions). Compared with the only other TD method applicable in continuous settings (C-learning), TD InfoNCE can achieve a comparable loss with $130\times$ less data ($7.7 \times 10^4$ vs $10^7$ transitions). Even compared with the strongest baseline (successor representations), which makes assumptions (tabular MDPs) that our method avoids, TD InfoNCE can achieve a comparable error rate with almost $20\times$ fewer samples ($5.2 \times 10^5$ vs $10^7$ transitions).

### 4.4 DOES TD INFONCE ENABLE OFF-POLICY REASONING?

The explicit temporal difference update (Eq. 10) in TD InfoNCE is similar to the standard Bellman backup, motivating us to study whether the resulting goal-conditioned policy is capable of performing dynamic programming with offline data. To answer these questions, we conduct two experiments on the same gridworld environment as in Sec. 4.3, comparing TD InfoNCE to contrastive RL (i.e., Monte Carlo InfoNCE). Fig. 4 shows that TD InfoNCE successfully stitches together pieces of different trajectories to find a route between unseen (state, goal) pairs. Fig. 5 shows that TD InfoNCE can perform off-policy reasoning, finding a path that is shorter than the average path demonstrated in the dataset. See Appendix D.4 for details.

## 5 CONCLUSION

This paper introduced a temporal difference estimator for the InfoNCE loss. Our goal-conditioned RL algorithm based on this estimator outperforms prior methods in both online and offline settings, and is capable of handling stochasticity in the environment dynamics. While we focused on a specific type of RL problem (goal-conditioned RL), in principle the TD InfoNCE estimator can be used to drive policy evaluation for arbitrary reward functions. One area for future work is to determine how it compares to prior off-policy evaluation techniques.

While we focused on evaluating the TD InfoNCE estimator on control tasks, it is worth noting that the MC InfoNCE objective has been previously applied to NLP, audio, video settings; one intriguing and important question is whether the benefits of TD learning seen on these control tasks translate into better representations in these other domains.

**Limitations.** One limitation of TD InfoNCE is complexity: compared with its Monte Carlo counterpart, ours is more complex and requires more hyperparameters. It is worth noting that even TD InfoNCE struggles to solve the most challenging control tasks with image observations. On the theoretical front, our convergence proof uses a slightly modified version of our loss (replacing a sum with an expectation), which would be good to resolve in future work.

**Acknowledgements** We thank Ravi Tej and Wenzhe Li for discussions about this work, and anonymous reviewers for providing feedback on early versions of this work. We thank Tongzhou Wang for providing performance of baselines in online GCRL experiments and thank Raj Ghugare for sharing code of environment implementation. We thank Vivek Myers for finding issues in the code. BE is supported by the Fannie and John Hertz Foundation.

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

## A   THEORETICAL ANALYSIS

Our convergence proof will focus on the tabular setting with known $p(s' \mid s, a)$ and $p(s_{t+})$, and follows the fitted Q-iteration strategy (Fu et al., 2019; Ernst et al., 2005; Bertsekas & Tsitsiklis, 1995): at each iteration, an optimization problem will be solved exactly to yield the next estimate of the discounted state occupancy measure. One key step in the proof is to employ a preserved invariant; we will define the classifier derived from the TD InfoNCE objective (Eq. 10) and show that this classifier always represents a valid probability distribution (over future states). We then construct a variant of the TD InfoNCE objective using this classifier and prove that optimizing this objective is exactly equivalent to perform policy evaluation, resulting in the convergence to the discounted state occupancy measure.

**Definition of the classifier.**   We start by defining the classifier derived from the TD InfoNCE as

$$C(s, a, s_{t+}) \triangleq \frac{p(s_{t+})e^{f(s,a,s_{t+})}}{\mathbb{E}_{p(s'_{t+})}\left[e^{f(s,a,s'_{t+})}\right]} = \frac{p(s_{t+})e^{f(s,a,s_{t+})}}{\sum_{s'_{t+} \in \mathcal{S}} p(s_{t+})e^{f(s,a,s'_{t+})}}, \tag{12}$$

suggesting that $C(s, a, \cdot)$ is a valid distribution over future states: $C(s, a, \cdot) \in \Delta(\mathcal{S})$.

**A variant of TD InfoNCE.**   Our definition of the classifier (Eq. 12) allows us to rewrite the importance weight $w(s, a, s_{t+})$ and softmax functions in $\mathcal{L}_{\text{TD InfoNCE}}$ (Eq. 10) as Monte Carlo estimates of the classifier using samples of $s_{t+}^{(1:N)}$,

$$w(s, a, s_{t+}^{(1:N)}) = \frac{e^{f(s,a,s_{t+}^{(1)})}}{\frac{1}{N}\sum_{i=1}^{N} e^{f(s,a,s_{t+}^{(i)})}} \approx \frac{C(s, a, s_{t+})}{p(s_{t+})}.$$

Thus, we construct a variant of the TD InfoNCE objective using $C$:

$$\bar{\mathcal{L}}_{\text{TD InfoNCE}}(C) \triangleq \mathbb{E}_{p(s,a)} \Big[ (1 - \gamma)\mathbb{E}_{p(s'=s_{t+}|s,a)} \left[ \log C(s, a, s_{t+}) \right]$$
$$+ \gamma \mathbb{E}_{\substack{p(s'|s,a),\pi(a'|s') \\ p(s_{t+})}} \left[ \frac{\lfloor C(s', a', s_{t+}) \rfloor_{\text{sg}}}{p(s_{t+})} \log C(s, a, s_{t+}) \right] \Big].$$

This objective is similar to $\mathcal{L}_{\text{TD InfoNCE}}$, but differs in that *(a)* softmax functions are replaced by $C(s, a, s_{t+})$ up to constant $\frac{1}{N \cdot p(s_{t+})}$ and *(b)* $w(s', a', s_{t+}^{(1:N)})$ is replaced by $\frac{C(s',a',s_{t+})}{p(s_{t+})}$. Formally, $\mathcal{L}_{\text{TD InfoNCE}}(C)$ is a nested Monte Carlo estimator of $\bar{\mathcal{L}}_{\text{TD InfoNCE}}$ (Rainforth et al., 2018; Giles, 2015) and we leave the analysis of the gap between them as future works. We now find the solution of $\bar{\mathcal{L}}_{\text{TD InfoNCE}}(C)$ analytically by rewriting it using the cross entropy and ignore the stop gradient operator to reduce clutter: $\bar{\mathcal{L}}_{\text{TD InfoNCE}}(C) =$

$$\mathbb{E}_{p(s,a)} \left[ (1 - \gamma)\mathbb{E}_{p(s'=s_{t+}|s,a)} \left[ \log C(s, a, s_{t+}) \right] + \gamma \mathbb{E}_{\substack{p(s'|s,a),\pi(a'|s',g) \\ C(s',a',s_{t+})}} \left[ \log C(s, a, s_{t+}) \right] \right]$$
$$= -\mathbb{E}_{p(s,a)} \left[ (1 - \gamma)\mathcal{CE} \left( p(s' = \cdot \mid s, a), C(s, a, \cdot) \right) \right.$$
$$\left. + \gamma \mathcal{CE} \left( \mathbb{E}_{p(s'|s,a),\pi(a'|s')} \left[ C(s', a', \cdot) \right], C(s, a, \cdot) \right) \right]$$
$$= -\mathbb{E}_{p(s,a)} \left[ \mathcal{CE} \left( (1 - \gamma)p(s' = \cdot \mid s, a) + \gamma \mathbb{E}_{p(s'|s,a),\pi(a'|s')} \left[ C(s', a', \cdot) \right], C(s, a, \cdot) \right) \right], \tag{13}$$

where the cross entropy for $p, q \in \Delta(\mathcal{X})$ is defined as

$$\mathcal{CE}(p(\cdot), q(\cdot)) = -\mathbb{E}_{p(x)}[\log q(x)] = -\sum_{x \in \mathcal{X}} p(x) \log q(x),$$

with the minimizer $q^\star = \arg\min_{q \in \Delta(\mathcal{X})} \mathcal{CE}(p(\cdot), q(\cdot)) = p$. Note that $p(s' = \cdot \mid s, a) \in \Delta(\mathcal{S})$ and $\mathbb{E}_{p(s'|s,a)\pi(a'|s')}[C(s', a', \cdot)] \in \Delta(\mathcal{S})$ in Eq. 13 indicate that their convex combination is also a distribution in $\Delta(\mathcal{S})$. This objective suggests a update for the classifier given any $(s, a, s_{t+})$:

$$C(s, a, s_{t+}) \leftarrow (1 - \gamma)p(s' = s_{t+} \mid s, a) + \gamma \mathbb{E}_{p(s'|s,a)\pi(a'|s')}[C(s', a', s_{t+})], \tag{14}$$

which bears a resemblance to the standard Bellman equation.

**InfoNCE Bellman operator.** We define the InfoNCE Bellman operator for any function $Q(s, a, s_{t+}) : \mathcal{S} \times \mathcal{A} \times \mathcal{S} \mapsto \mathbb{R}$ with policy $\pi(a \mid s)$ as

$$\mathcal{T}_{\text{InfoNCE}} Q(s, a, s_{t+}) \triangleq (1 - \gamma) p(s' = s_{t+} \mid s, a) + \gamma \mathbb{E}_{p(s' \mid s, a) \pi(a' \mid s')}[Q(s', a', s_{t+})], \quad (15)$$

and write the update of the classifier as $C(s, a, s_{t+}) \leftarrow \mathcal{T}_{\text{InfoNCE}} C(s, a, s_{t+})$. Like the standard Bellman operator, this InfoNCE Bellman operator is a $\gamma$-contraction. Unlike the standard Bellman operator, $\mathcal{T}_{\text{InfoNCE}}$ replaces the reward function with the discounted probability of the future state being the next state $(1 - \gamma) p(s' = s_{t+} \mid s, a)$ and applies to a function depending on a state-action pair and a future state $(s, a, s_{t+})$.

**Proof of convergence.** Using the same proof of convergence for policy evaluation with the standard Bellman equation (Sutton & Barto, 2018; Agarwal et al., 2019), we conclude that repeatedly applying $\mathcal{T}_{\text{InfoNCE}}$ to $C$ results in convergence to a unique $C^\star$ regardless of initialization,

$$C^\star(s, a, s_{t+}) = (1 - \gamma) p(s' = s_{t+} \mid s, a) + \gamma \mathbb{E}_{p(s' \mid s, a) \pi(a' \mid s')}[C^\star(s', a', s_{t+})].$$

Since $C^\star(s, a, s_{t+})$ and $p^\pi(s_{t+} \mid s, a)$ satisfy the same identity (Eq. 4), we have $C^\star(s, a, s_{t+}) = p^\pi(s_{t+} \mid s, a)$, i.e., the classifier of the TD InfoNCE estimator converges to the discounted state occupancy measure. To recover $f^\star$ from $C^\star$, we note that $f^\star$ satisfies

$$f^\star(s, a, s_{t+}) = \log C^\star(s, a, s_{t+}) - \log p(s_{t+}) + \log \mathbb{E}_{p(s'_{t+})}[\exp(f^\star(s, a, s'_{t+}))]$$

$$= \log p^\pi(s_{t+} \mid s, a) - \log p(s_{t+}) + \log \mathbb{E}_{p(s'_{t+})}[\exp(f^\star(s, a, s'_{t+}))]$$

by definition. Since the (expected) softmax function is invariant to translation, we can write $f^\star(s, a, s_{t+}) = \log p^\pi(s_{t+} \mid s, a) - \log p(s_{t+}) - \log c(s, a)$, where $c(s, a)$ is an arbitrary function that does not depend on $s_{t+}$ [2]. Thus, we conclude that TD InfoNCE objective converges to the same solution as that of MC InfoNCE (Eq. 7), i.e. $\tilde{\mathcal{L}}_{\text{TD InfoNCE}}(f^\star) = \mathcal{L}_{\text{MC InfoNCE}}(f^\star)$.

It is worth noting that the same proof applies to the goal-conditioned TD InfoNCE objective. After finding an exact estimate of the discounted state occupancy measure of a goal-conditioned policy $\pi(a \mid s, g)$, maximizing the policy objective (Eq. 11) is equivalent to doing policy improvement. We can apply the same proof as in the Lemma 5 of (Eysenbach et al., 2020) to conclude that $\pi(a \mid s, g)$ converges to the optimal goal-conditioned policy $\pi^\star(a \mid s, g)$.

## B  CONNECTION WITH MUTUAL INFORMATION AND SKILL LEARNING.

The theoretical analysis in Appendix A has shown that the TD InfoNCE estimator has the same effect as the MC InfoNCE estimator. As the (MC) InfoNCE objective corresponds to a lower bound on mutual information (Poole et al., 2019), we can interpret our goal-conditioned RL method as having both the actor and the critic jointly optimize a lower bound on mutual information. This perspective highlights the close connection between unsupervised skill learning algorithms (Eysenbach et al., 2018; Campos et al., 2020; Warde-Farley et al., 2018; Gregor et al., 2016), and goal-conditioned RL, a connection previously noted in Choi et al. (2021). Seen as an unsupervised skill learning algorithm, goal-conditioned RL lifts one of the primary limitations of prior methods: it can be unclear which skill will produce which behavior. In contrast, goal-conditioned RL methods learn skills that are defined as optimizing the likelihood of reaching particular goal states.

## C  CONNECTION WITH SUCCESSOR REPRESENTATIONS

In settings with tabular states, the successor representation (Dayan, 1993) is a canonical method for estimating the discounted state occupancy measure (Eq. 3). The successor representation has strong ties to cognitive science (Gershman, 2018) and has been used to accelerate modern RL methods (Barreto et al., 2017; Touati & Ollivier, 2021).

---

[2]Technically, $f^\star$ should be a set of functions satisfying $\left\{ f : \dfrac{e^{f(s, a, s_{t+})}}{\mathbb{E}_{p(s'_{t+})}\left[e^{f(s, a, s'_{t+})}\right]} = \dfrac{C^\star(s, a, s_{t+})}{p(s_{t+})} \right\}.$

Successor representation $M^\pi : \mathcal{S} \times \mathcal{A} \mapsto \Delta(\mathcal{S})$ is a long-horizon, policy dependent model that estimates the discounted state occupancy measure for every $s \in \mathcal{S}$ via the recursive relationship (Eq. 4). Given a policy $\pi(a \mid s)$, the successor representation satisfies

$$M^\pi(s, a) \leftarrow (1 - \gamma)\text{ONEHOT}_{|\mathcal{S}|}(s') + \gamma M^\pi(s', a'),  \tag{16}$$

where $s' \sim p(s' \mid s, a)$ and $a' \sim \pi(a' \mid s')$. Comparing this update to the TD InfoNCE update shown in Fig. 1 and Eq. 14, we see that this successor representation update is a special case of TD InfoNCE where *(a)* every state is used instead of randomly-sampling the states, and *(b)* the probabilities are encoded directed in a matrix $M$, rather than encoding the probabilities as the inner product between two learned vectors.

This connection is useful because it highlights how and why the learned representations can be used to solve fully-general reinforcement learning tasks. In the same way that the successor representation can be used to express the value function of a reward ($M^\pi(s, a)^\top r(\cdot)$), the representations learned by TD InfoNCE can be used to recover value functions:

$$\hat{Q}^\pi(s, a) = r(s, a) + \frac{\gamma}{1 - \gamma}\mathbb{E}_{s_{t+}^{(1:N)} \sim p(s_{t+}), a_{t+} \sim \pi(a|s_{t+}^{(1)})}\left[\frac{e^{f(s, a, s_{t+}^{(1)})}}{\frac{1}{N}\sum_{i=1}^{N} e^{f(s, a, s_{t+}^{(i)})}} r(s_{t+}^{(1)}, a_{t+})\right]$$

See Mazoure et al. (2022) for details on this construction.

## D    EXPERIMENTAL DETAILS

### D.1    THE COMPLETE ALGORITHM FOR GOAL-CONDITIONED RL

The complete algorithm of TD InfoNCE alters between estimating the discounted state occupancy measure of the current goal-conditioned policy via contrastive learning (Eq. 10) and updating the policy using the actor loss (Eq. 11), while collecting more data. Given a batch of $N$ transitions of $\{(s_t^{(i)}, a_t^{(i)}, s_{t+1}^{(i)}, g^{(i)}, s_{t+}^{(i)})\}_{i=1}^{N}$ sampled from $p(s_t, a_t, g)$, $p(s_{t+1} \mid s_t, a_t)$, and $p(s_{t+})$, we can first compute the critic function for different combinations of goal-conditioned state-action pairs and future states by computing their contrastive representations $\phi(s_t, a_t, g)$, $\psi(s_{t+})$, and $\psi(s_{t+})$, and then construct two critic matrices $F_{\text{next}}, F_{\text{future}} \in \mathbb{R}^{N \times N}$ using those representations:

$$F_{\text{next}}[i, j] = \phi(s_t^{(i)}, a_t^{(i)}, g^{(i)})^\top \psi(s_{t+1}^{(j)}), F_{\text{future}}[i, j] = \phi(s_t^{(i)}, a_t^{(i)}, g^{(i)})^\top \psi(s_{t+}^{(j)})$$

Note that the inner product parameterization of the critic function $f(s_t, a_t, g, s_{t+}) = \phi(s_t, a_t, g)^\top \psi(s_{t+})$ helps compute these matrices efficiently. Using these critic matrices, we rewrite the TD InfoNCE estimate as a sum of two cross entropy losses. The first cross entropy loss involves predicting which of the $N$ next states $s_{t+1}^{(1:N)}$ is the correct next state for the corresponding goal-conditioned state and action pair:

$$(1 - \gamma)\mathcal{CE}(\text{logits} = F_{\text{next}}, \text{labels} = I_N),$$

where $\mathcal{CE}(\text{logits} = F_{\text{next}}, \text{labels} = I_N) = -\sum_{i=1}^{N}\sum_{j=1}^{N} I_N[i, j] \cdot \log \text{SOFTMAX}(F_{\text{next}})[i, j]$, $\text{SOFTMAX}(\cdot)$ denotes row-wise softmax normalization, and $I_N$ is a $N$ dimensional identity matrix. For the second cross entropy term, we first sample a batch of $N$ actions from the target policy at the *next* time step, $a_{t+1}^{(1:N)} \sim \pi(a_{t+1} \mid s_{t+1}, g)$, and then estimate the importance weight matrix $W \in \mathbb{R}^{N \times N}$ that serves as labels as

$$F_w[i, j] = \phi(s_{t+1}^{(i)}, a_{t+1}^{(i)}, g^{(i)})^\top \psi(s_{t+}^{(j)}), W = N \cdot \text{SOFTMAX}(F_w).$$

Thus, the second cross entropy loss takes as inputs the critic $F_{\text{future}}$ and the importance weight $W$:

$$\gamma\mathcal{CE}(\text{logits} = F_{\text{future}}, \text{labels} = W).  \tag{17}$$

Regarding the policy objective (Eq. 11), it can also be rewritten as the cross entropy between a critic matrix $F_{\text{goal}}$ with $F_{\text{goal}}[i, j] = \phi(s_t^{(i)}, a^{(i)}, g^{(i)})^\top \psi(g^{(j)})$, where $a^{(i)} \sim \pi(a \mid s_t^{(i)}, g^{(i)})$, and the identity matrix $I_N$:

$$\mathcal{CE}(\text{logits} = F_{\text{goal}}, \text{labels} = I_N)$$

In practice, we use neural networks with parameters $\theta = \{\theta_\phi, \theta_\psi\}$ to parameterize (normalized) contrastive representations $\phi$ and $\psi$ and use a neural network with parameters $\omega$ to parameterize the goal-conditioned policy $\pi$ and optimize them using gradient descent.

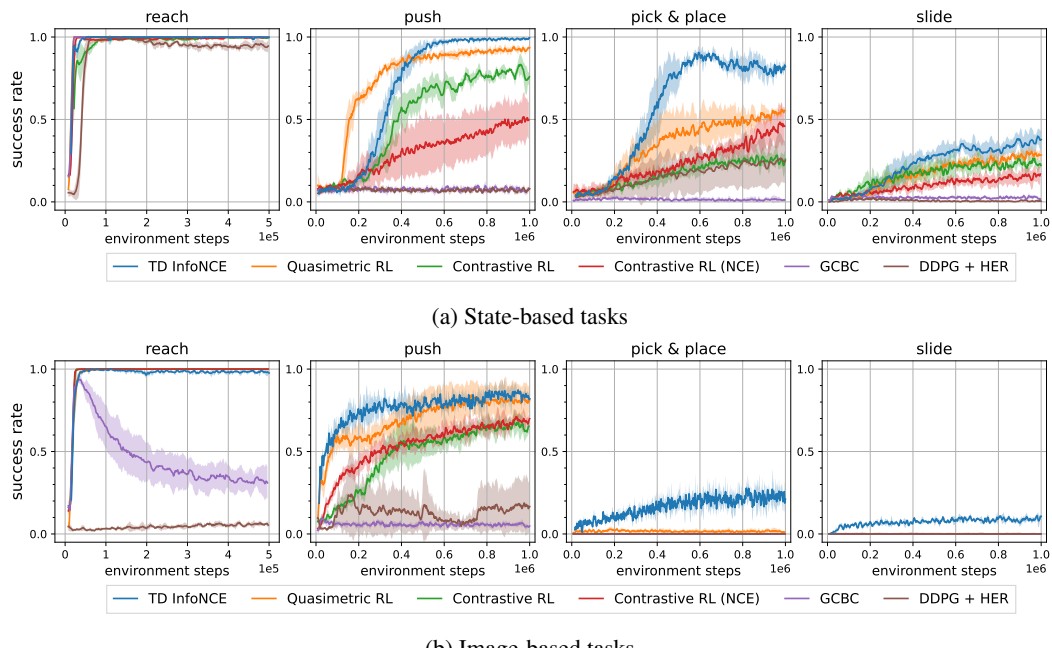

(a) State-based tasks

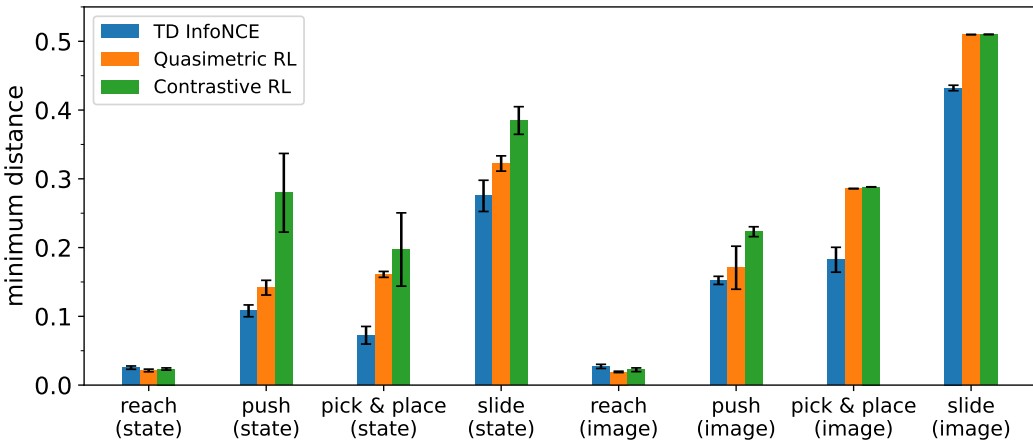

(b) Image-based tasks

Figure 6: **Evaluation on online GCRL benchmarks.** TD InfoNCE matches or outperforms all baselines on both state-based and image-based tasks.

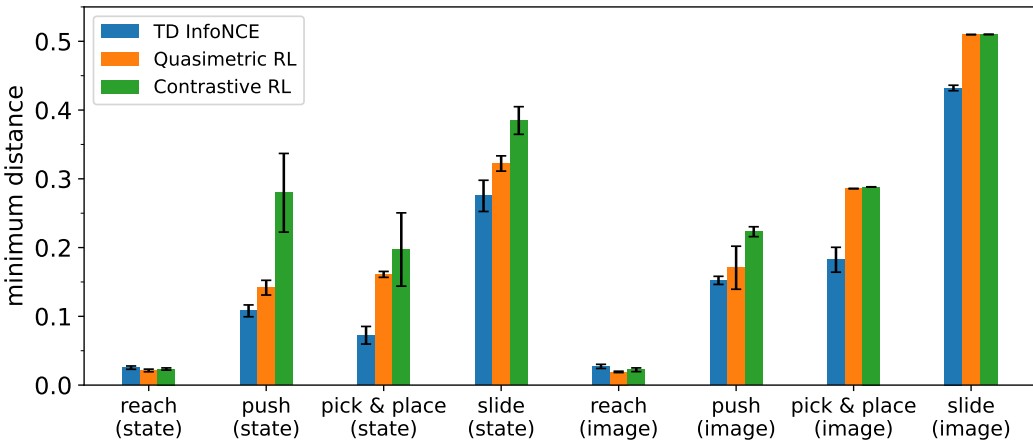

Figure 7: We also compare different methods using the minimum distance of the gripper or the object to the goal over an episode. Note that a lower minimum distance indicates a better performance. TD InfoNCE achieves competitive minimum distances on online GCRL benchmarks.

## D.2 ONLINE GOAL-CONDITIONED RL EXPERIMENTS

We report complete success rates for online GCRL experiments in Fig. 6, showing the mean success rate and standard deviation (shaded regions) across five random seeds. TD InfoNCE outperforms or achieves similar performance on all tasks, compared with other baselines. For those tasks where the success rate fails to separate different methods significantly (e.g., `slide (state)` and `push (image)`), we include comparisons using minimum distances of the gripper or the object to the goal over an episode in Fig. 7, selecting the strongest baselines QRL and contrastive RL. Note that a lower minimum distance indicates a better performance. These results suggest that TD InfoNCE is able to emerge a goal-conditioned policy by estimating the discounted state occupancy measure, serving as a competitive goal-conditioned RL algorithm.

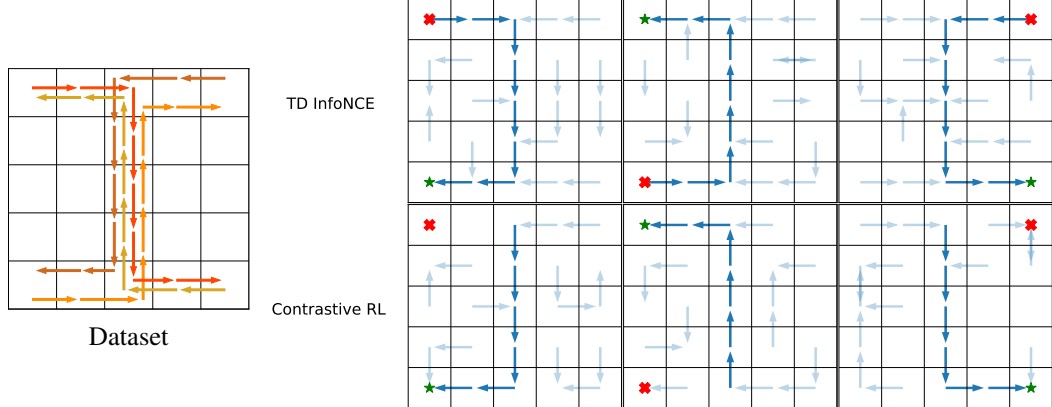

Figure 8: **Stitching trajectories in a dataset.** We show additional (start, goal) pairs for the experiment in Fig. 4.

### D.3 OFFLINE GOAL-CONDITIONED RL EXPERIMENTS

Similar to prior works (Eysenbach et al., 2022; Wang et al., 2023), we adopt an additional goal-conditioned behavioral cloning regularization to prevent the policy from sampling out-of-distribution actions (Fujimoto & Gu, 2021; Kumar et al., 2020; 2019) during policy optimization (Eq.5):

$$\underset{\pi(\cdot|\cdot,\cdot)}{\arg\max} \, \mathbb{E}_{\substack{(s,a_{\mathrm{orig}},g)\sim p(s,a_{\mathrm{orig}},g) \\ a\sim\pi(a|s,g),s_{t+}^{(1:N)}\sim p(s_{t+})}} \left[ (1-\lambda)\cdot\log\frac{e^{f(s,a,g,s_{t+}=g)}}{\sum_{i=1}^{N}e^{f(s,a,g,s_{t+}^{(i)})}} + \lambda\cdot\|a-a_{\mathrm{orig}}\|_2^2 \right],$$

where $\lambda$ is the coefficient for regularization. Note that we use a supervised loss based on the mean squared error instead of the maximum likelihood estimate of $a_{\mathrm{orig}}$ under policy $\pi$ used in prior works. We compare TD InfoNCE to the state-of-the-art QRL (Wang et al., 2023) and its Monte Carlo counterpart (contrastive RL (Eysenbach et al., 2022)). We also compare to the pure goal-conditioned behavioral cloning implemented in (Emmons et al., 2021) as well as a recent baseline that predicts optimal actions via sequence modeling using a transformer (DT (Chen et al., 2021)). Our last two baselines are offline actor-critic methods trained via TD learning: TD3 + BC (Fujimoto & Gu, 2021) and IQL (Kostrikov et al., 2021), not involving goal-conditioned relabeling. We use the result for baselines except QRL from (Eysenbach et al., 2022).

As shown in Table 1, TD InfoNCE matches or outperforms all baselines on 5 / 6 tasks. On tasks (`medium-play-v2` and `medium-diverse-v2`), TD InfoNCE achieves a $+13\%$ improvement over contrastive RL, showing the advantage of temporal difference learning over the Monte Carlo approach with a fixed dataset. We conjecture that this benefit comes from the dynamic programming property of the TD method and will investigate this property further in later experiments (Sec. 4.4). Additionally, TD InfoNCE performs $1.4\times$ better than GCBC and retains a $3.8\times$ higher scores than DT on average, where these baselines use (autoregressive) supervised losses instead of TD learning. These results suggest that TD InfoNCE is also a competitive goal-conditioned RL algorithm in the offline setting.

### D.4 OFF-POLICY REASONING EXPERIMENTS

**Stitching trajectories.** The first set of experiments investigate whether TD InfoNCE successfully stitches pieces of trajectories in a dataset to find complete paths between (start, goal) pairs unseen together in the dataset. We collect a dataset with size 20K consisting of "Z" style trajectories moving in diagonal and off-diagonal directions (Fig. 8), while evaluating the learned policy on reaching goals on the same edge as starting states after training both methods for 50K gradient steps. Figure 8 shows that TD InfoNCE succeeds in stitching parts of trajectory in the dataset, moving along "C" style paths towards goals, while contrastive RL fails to do so. These results justify our hypothesis that TD InfoNCE performs dynamic programming and contrastive RL instead naively follows the behavior defined by the data.

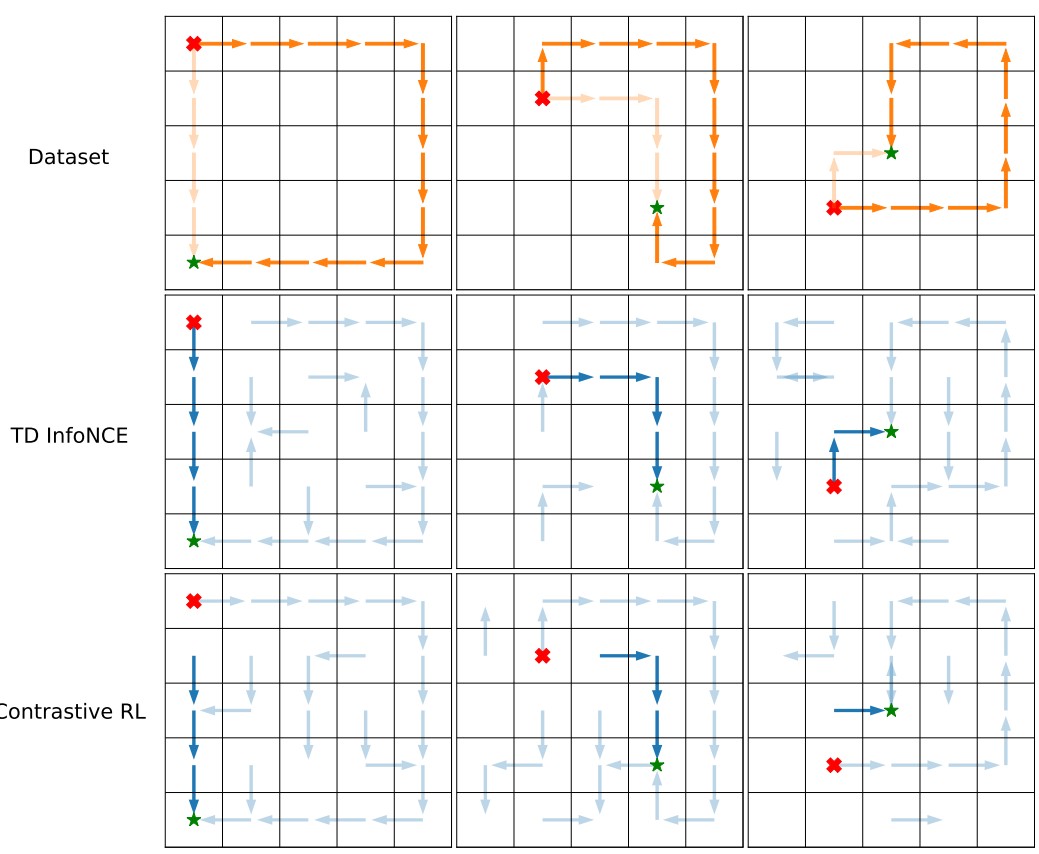

Figure 9: **Searching for shortcuts in skewed datasets.** We show additional (start, goal) pairs for the experiment in Fig. 5.

Table 2: Hyperparameters for TD InfoNCE.

| Hyperparameters | Values |
|---|---|
| actor learning rate | $5 \times 10^{-5}$ |
| critic learning rate | $3 \times 10^{-4}$ |
| using $\ell_2$ normalized representations | yes |
| hidden layers sizes (for both actor and representations) | $(512, 512, 512, 512)$ |
| contrastive representation dimensions | 16 |

Table 3: Changes to hyperparameters for offline RL experiments. (Table 1)

| Hyperparameters | Values |
|---|---|
| batch size (on `large-` tasks) | $256 \rightarrow 1024$ |
| hidden layers sizes (for both actor and representations on `large-` tasks) | $(512, 512, 512, 512) \rightarrow (1024, 1024, 1024, 1024)$ |
| behavioral cloning regularizer coefficient $\lambda$ | 0.1 |
| goals for actor loss | random states $\rightarrow$ future states |

**Searching for shortcuts.**   Our second set of experiments aim to compare the performance of TD InfoNCE against contrastive RL on searching shortcuts in skewed datasets. To study this, we collect different datasets of size 20K with trajectories conditioned on the same pair of initial state and goal, with $95\%$ of the time along a long path and $5\%$ of the time along a short path. Using these skewed datasets, we again train both methods for 50K gradient steps and then evaluate the policy performance on reaching the same goal starting from the same state. We show the goal-conditioned policies learned by the two approaches in Fig. 9. The observation that TD InfoNCE learns to take shortcuts even though those data are rarely seen, while contrastive RL follows the long paths dominating the entire dataset, demonstrates the advantage of temporal difference learning over its Monte Carlo counterpart in improving data efficiency.

## D.5   Implementations and Hyperparameters

We implement TD InfoNCE, contrastive RL, and C-Learning using JAX (Bradbury et al., 2018) building upon the official codebase of contrastive RL[3]. For baselines QRL, GCBC, and DDPG + HER, we use implementation provided by the author of QRL[4]. We summarize hyperparameters for TD InfoNCE in Table 2. Whenever possible, we used the same hyperparameters as contrastive RL (Eysenbach et al., 2022). Since TD InfoNCE computes the loss with $N^2$ negative examples, we increase the capacity of the goal-conditioned state-action encoder and the future state encoder to 4 layers MLP with 512 units in each layer applying ReLU activations. For fair comparisons, we also increased the neural network capacity in baselines to the same number and used a fixed batch size 256 for all methods. Appendix E.1 includes ablations studying the effect of differet hyperparamters in Table 2. For offline RL experiments, we make some changes to hyperparameters (Table 3).

## E   Additional Experiments

### E.1   Hyperparameter Ablations

We conduct ablation experiments to study the effect of different hyperparameters in Table 2, aiming to find the best hyperparameters for TD InfoNCE. For each hyperparameter, we selected a set of

---

[3]https://github.com/google-research/google-research/tree/master/contrastive_rl
[4]https://github.com/quasimetric-learning/quasimetric-rl

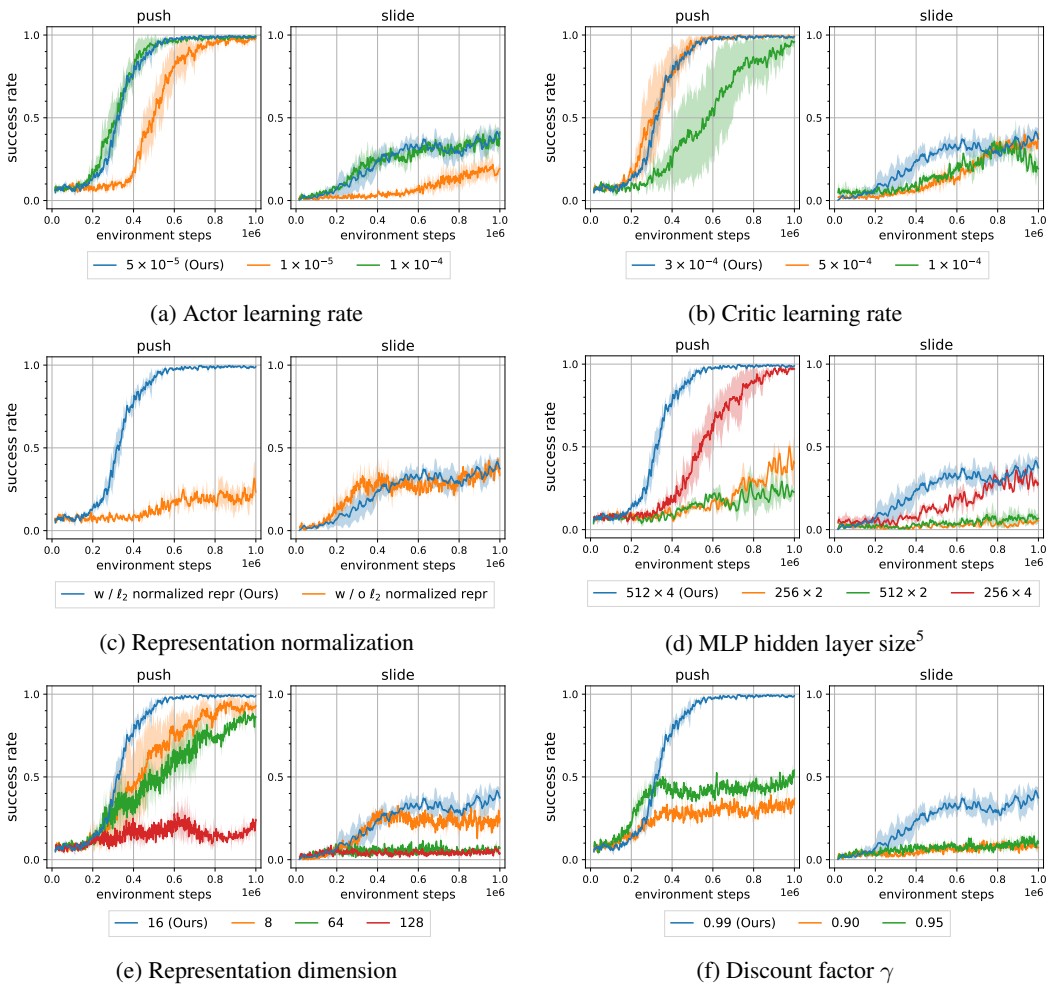

Figure 10: **Hyperparameter ablation.** We conduct ablations to study the effect of different hyper-paramters listed in Table 2 and the discout factor $\gamma$ on state-based `push` and `slide`.

different values and conducted experiments on `push (state)` and `slide (state)`, one easy task and one challenging task, for five random seeds. We report mean and standard deviation of success rates in Fig. 10. These results suggest that while some values of the hyperparameter have similar effects, e.g. actor learning rate $= 5 \times 10^{-5}$ vs $1 \times 10^{-4}$, our choice of combination is optimal for TD InfoNCE.

### E.2 PREDICTING THE DISCOUNTED STATE OCCUPANCY MEASURE

Our experiments estimating the discounted state occupancy measure in the tabular setting (Sec. 4.3) observed a small "irreducible" error. To test the correctness of our implementation, we applied the successor representation with a known model (Fig. 11), finding that its error does go to zero. This gives us confidence that our implementation of the successor representation baseline is correct, and suggests that the error observed in Fig. 3 arises from sampling the transitions (rather than having a known model).

### E.3 UNDERSTANDING THE DIFFERENCES BETWEEN TD INFONCE AND C-LEARNING

While conceptually similar, our method is a temporal difference estimator building upon InfoNCE whereas C-learning instead bases on the NCE objective (Gutmann & Hyvärinen, 2010). There are

---

[5]We use $x \times y$ to denote a $y$ layers MLP with $x$ units in each layer.

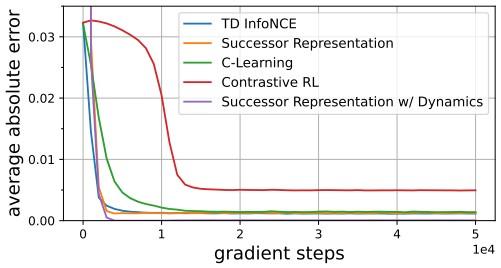

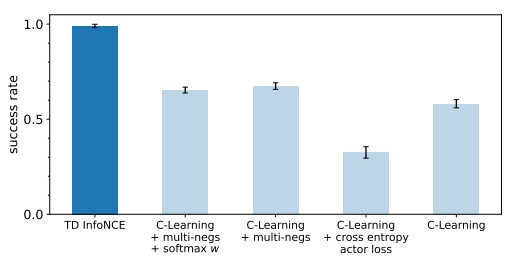

Figure 11: Errors of discounted state occupancy measure estimation in a tabular setting.

Figure 12: Differences between TD InfoNCE and C-Learning.

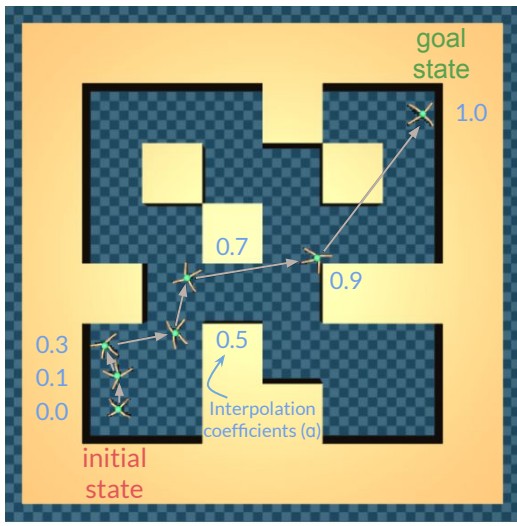

(a) Parametric interpolation

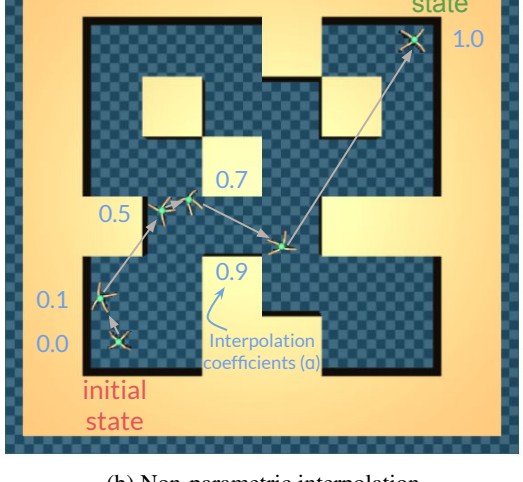

(b) Non-parametric interpolation

Figure 13: **Visualizing representation interpolation.** Using spherical interpolation of representations *(Left)* or linear interpolation of softmax features *(Right)*, TD InfoNCE learns representations that capture not only the content of states, but also the causal relationships.

mainly three distinctions between TD InfoNCE and C-Learning: *(a)* C-Learning uses a binary cross entropy loss, while TD InfoNCE uses a categorical cross entropy loss. *(b)* C-Learning uses importance weights of the form $\exp(f(s, a, g))$; if these weights are self-normalized (Dubi & Horowitz, 1979; Hammersley, 1956), they corresponds to the softmax importance weights in our objectives (Eq. 9). *(c)* For the same batch of $N$ transitions, TD InfoNCE updates representations of $N^2$ negative examples (Eq. 17), while C-Learning only involves $N$ negative examples. We ablate these decisions in Fig. 12, finding that differences (b) and (c) have little effect. Thus, we attribute the better performance of TD InfoNCE to its use of the categorical cross entropy loss.

### E.4 REPRESENTATION INTERPOLATION

Prior works have shown that representations learned by self-supervised learning incorporate structure of the data (Wang & Isola, 2020; Arora et al., 2019), motivating us to study whether the representations acquired by TD InfoNCE contain task-specific information. To answer this question, we visualize representations learned by TD InfoNCE via interpolating in the latent space following prior work (Zheng et al., 2023). We choose to interpolate representations learned on the offline AntMaze `medium-play-v2` task and compare a parametric interpolation method against a non-parametric variant. Importantly, the states and goals of this task are 29 dimensions and we visualize them in 2D from a top-down view.

**Parametric interpolation.** Given a pair of start state and goal $(s_0, g)$, we compute the normalized representations $\phi(s_0, a_{\text{no-op}}, g)$ and $\phi(g, a_{\text{no-op}}, g)$, where $a_{\text{no-op}}$ is an action taking no operation. Applying spherical linear interpolation to both of them results in blended representations,

$$\frac{\sin(1-\alpha)\eta}{\sin\eta}\phi(s_0, a_{\text{no-op}}, g) + \frac{\sin\alpha\eta}{\sin\eta}\phi(g, a_{\text{no-op}}, a),$$

where $\alpha \in [0, 1]$ is the interpolation coefficient and $\eta$ is the angle subtended by the arc between $\phi(s_0, a_{\text{no-op}}, g)$ and $\phi(g, a_{\text{no-op}}, g)$. These interpolated representations can be used to find the spherical nearest neighbors in a set of representations of validation states $\{\phi(s_{\text{val}}, a_{\text{no-op}}, g)\}$ and we call this method parametric interpolation.

**Non-parametric interpolation.** We can also sample another set of random states and using their representations $\{\phi(s_{\text{rand}}^{(i)}, a_{\text{no-op}}, g)\}_{i=1}^{S}$ as anchors to construct a softmax feature for a state $s$, $\text{feat}(s; g, \{s_{\text{rand}}\})$:

$$\text{SOFTMAX}\left(\left[\phi(s, a_{\text{no-op}}, g)^{\top}\phi(s_{\text{rand}}^{(1)}, a_{\text{no-op}}, g), \cdots, \phi(s, a_{\text{no-op}}, g)^{\top}\phi(s_{\text{rand}}^{(S)}, a_{\text{no-op}}, g)\right]\right).$$

We compute the softmax features for representations of start and goal states and then construct the linear interpolated features,

$$\alpha\text{feat}(s_0; g, \{s_{\text{rand}}\}) + (1-\alpha)\text{feat}(g; g, \{s_{\text{rand}}\}).$$

Those softmax features of interpolated representations are used to find the $\ell_2$ nearest neighbors in a softmax feature validation set. We call this method non-parametric interpolation.

Results in Fig. 13 suggest that when interpolating the representations using both methods, the intermediate representations correspond to sequences of states that the optimal policy should visit when reaching desired goals. Therefore, we conjecture that TD InfoNCE encodes causality in its representations while the policy learns to arrange them in a temporally correct order.

