# OpenReview forum: "Contrastive Difference Predictive Coding"
_ICLR.cc/2024/Conference — ICLR 2024 poster_

### Official Review · Reviewer_Jmo8 · 2023-10-26

**Soundness:** 3 good
**Presentation:** 3 good
**Contribution:** 2 fair
**Rating:** 6
**Confidence:** 4

**Summary:**

This paper studies the representation learning for goal-conditioned reinforcement learning problems. Built upon InfoNCE objective, this paper proposes a temporal difference estimator of InfoNCE objective and applied it to goal-conditioned RL algorithm.

In the experiment section, in the online RL setting, the proposed method is compared with prior goal-conditioned RL algorithms, including quasimetric RL, contrastive RL (Monte Carlo estimator of NCE objective), and hindsight experience relabelling. The proposed method achieved a higher average return in the comparison. Also, in the offline RL setting, The proposed method is compared with quasimetric RL, contrastive RL, and SOTA offline RL algorithms. The proposed algorithm generally outperforms baselines.

**Strengths:**

This paper is well-written and easy to read. The proposed method is explained and presented clearly.

This paper provides clear derivation and a solid theoretical foundation for the proposed method in Section 3. I

The proposed method is supported by extensive experiments comparing with many baseline approaches.

The analysis in Section 4.3 and 4.4 validate the advantages of the proposed method in comparison with other representation learning approaches. It is impressive to see that the proposed method can stitch together pieces of data.

**Weaknesses:**

Some statements, especially some in the introduction part, seem not fully supported by evidence provided in the paper.

For example, it is claimed that the proposed method "enables us to perform counterfactual reasoning". However, this point is not clear in the following section. Could you please explain it in detail?

**Questions:**

How is the proposed method sensitive to hyper-parameters? Do we need careful hyper-parameter tuning to make it work? Is there any intuitive guidance about how to adjust the hyper-parameters?

In Algorithm 1, many notations are introduced for the first time without any definition. Could you please clarify them?

One important baseline, contrastive RL, is the Monte Carlo estimator of the NCE loss. Could you please also compare with the algorithm using Monte Carlo estimator of InfoNCE loss, since it is already introduced in the prior work Eysenbach et al., 2022

---

> ### Author Response · Authors · 2023-11-19
> **Rebuttal by Authors**
>
> We thank the reviewer for the responses and suggestions for improving the paper. The reviewer brought up two main questions about hyperparameter selection and comparisons to Monte Carlo InfoNCE loss (contrastive RL (CPC)) in the experiments. We have attempted to address these questions by ablating different hyperparameters (new Fig. 10) and adding comparisons to contrastive RL (CPC) in online experiments (Fig. 2 and Appendix Fig. 6). **Together with the discussion below, does this fully address the reviewer’s concerns?**
>
> > How is the proposed method sensitive to hyper-parameters? Do we need careful hyper-parameter tuning to make it work? Is there any intuitive guidance about how to adjust the hyper-parameters?
>
> We ran additional ablation experiments to study the effect of different hyperparameters for TD InfoNCE listed in table 2 and the discount factor $\gamma$. For each hyperparameter, we selected a set of different values and conducted experiments on $\texttt{push (state)}$ and $\texttt{slide (state)}$, one easy task and one challenging task, for five random seeds. We report mean and standard deviation of success rates in Appendix E.1. These results suggest that while some values of the hyperparameter have similar effects, e.g. actor learning rate = $5 \times 10^{-5}$ vs $1 \times 10^{-4}$, our choice of combination is optimal for TD InfoNCE. For fair comparison, we fixed the batch size of TD InfoNCE to 256, the same number as other baselines (Appendix E of [1] and Appendix C.3 of [2]).
>
> > Could you please also compare with the algorithm using Monte Carlo estimator of InfoNCE loss
>
> As suggested by the reviewer, we ran new experiments comparing TD InfoNCE to contrastive RL (CPC) (MC InfoNCE) using the official contrastive RL repository [1]. We report results on the Fetch robotics benchmark in Fig. 2 and Appendix Fig. 6, showing mean and standard deviations over five random seeds. Note that we use contrastive RL to denote MC InfoNCE (contrastive RL (CPC)) and contrastive RL (NCE) to denote the original contrastive RL algorithm (See [1] for details). These results suggest that using InfoNCE loss for contrastive RL achieves a slightly better performance than using NCE loss for it on 4 / 8 tasks. Nonetheless, TD InfoNCE still outperforms this InfoNCE variant of contrastive RL (or Monte Carlo InfoNCE) and our conclusions remain the same. We have revised Sec. 4.1 to add this baseline.
>
> > For example, it is claimed that the proposed method "enables us to perform counterfactual reasoning". However, this point is not clear in the following section. Could you please explain it in detail?
>
> We have revised this sentence to clarify what we mean by counterfactual reasoning (off-policy reasoning): (a) successfully stitching together pieces of different trajectories in an offline dataset to find a path between unseen state and goal pairs; (b) Finding a path that is shorter than the average paths demonstrated in the dataset. These two capabilities are demonstrated by our experiments in Sec. 4.4 and Appendix D.4.
>
> > In Algorithm 1, many notations are introduced for the first time without any definition. Could you please clarify them?
>
> We have added notations in Algorithm 1 and defer details of the full goal-conditioned TD InfoNCE algorithm to Appendix D.1 due to limited pages.

---

> ### Comment · Reviewer_Jmo8 · 2023-11-20
> **Thanks for your detailed response**
>
> Thank authors for the detailed response. I especially appreciate the additional experiment results, which help mitigate my concerns. I'd keep leaning toward acceptance.

---

### Official Review · Reviewer_a2XQ · 2023-10-31

**Soundness:** 3 good
**Presentation:** 2 fair
**Contribution:** 3 good
**Rating:** 8
**Confidence:** 3

**Summary:**

This paper extends previous works on contrastive RL by using a temporal difference estimator. Similar to contrastive RL, the paper proposes to use contrastive learning, InfoNCE in particular, for estimating the discounted state occupancy measure. Unlike the previous contrastive RL approach, which averages over the goal distribution, the proposed method uses a temporal difference estimator and results in a Bellman-like update rule (TDInfoNCE). Comparing the Bellman update for value function, TDInfoNCE requires taking expectation over the future states from a potentially different goal. It turns out that this can be estimated using importance weight. Then, the paper shows how the estimated state occupancy measure can be used in conjuction with goal-conditioned RL to form a full-fledged RL agent. Experimental results are shown on both online and offline settings and compared to a range of existing methods. In both settings, TDInfoNCE outperforms in most of the environments compared to previous approaches.

**Strengths:**

The paper is mostly well written, apart from some details (see questions section).

The derivations are sound.

Experimental results show strong performance comparing to previous methods. The paper also presents some analysis and insights to explain the performance.

**Weaknesses:**

The novelty is slightly limited. The idea of using InfoNCE to estimate the state occupancy measure has been presented in contrastive RL; the Bellman-like update and the use of importance weight has been presented in C-Learning.

**Questions:**

1. Questions regarding the algorithm:
 - It's not clear to me what's the goal distribution is used? Is it a random goal? Does the different goal distribution affect data efficiency?
- How is $s_{t+}$ sampled? Is it the same as previous approaches - sample t from a geometric distribution?

2. The notation in Figure 1 is not very clear to me. Is it suppose to visualize Eq. 4?

3. In Eq. 7 and the one above, shoud it be $p^{\pi}(s_{t+}^{(1)}|s^{'}, a^{'}, g)$ instead of  $p^{\pi}(s_{t+}^{(1)}|s^{'}, a^{'})$?

4. In Table 1, result for contrastive RL on large-diverse-v2 should not be bold; result for TDInfoNCE on unmaze-v2 should not be bold.

---

> ### Author Response · Authors · 2023-11-19
> **Rebuttal by Authors**
>
> We thank the reviewer for the detailed response and helpful suggestions for improving the paper. It seems that the reviewer’s main question is about the novelty of the paper. Our work does build upon prior work of contrastive RL and C-Learning. Contrastive RL is an on-policy Monte Carlo method, while our proposed method is a off-policy temporal difference version of it, achieving higher sample efficiency. C-learning can be interpreted as a TD version of a NCE (2-class) contrastive loss, while our proposed method is a TD version of the infoNCE (many-class) contrastive loss. We have clarified this difference in the final paragraph of Sec. 3.1. Empirically, TD InfoNCE is $1500 \times$ more sample efficient than its Monte Carlo counterpart ($6.5 × 10^3$ vs $10^7$ transitions) and achieves a comparable loss with $130\times$ less data than C-Learning ($7.7 × 10^4$ vs $10^7$ transitions) (See Fig. 3 (Right)).  **Together with the discussion below, does this address the reviewer’s concerns?**
>
> > It's not clear to me what's the goal distribution is used? Is it a random goal? Does the different goal distribution affect data efficiency?
>
> We assume that the goal space is the same as the state space and do not constraint the distribution of goal. In practice, we usually define goals using human preference. For example, in task $\text{pick \\& place}$, we sample the target position of the robot arm to be either in the air or on the table with the block in the gripper.
>
> > How is s_{t+} sampled? Is it the same as previous approaches - sample t from a geometric distribution?
>
> Our TD InfoNCE is an off-policy algorithm, indicating that it does not require sample future states from an on-policy geometric distribution. Instead, we sample $s_{t+}$ randomly from the replay buffer or the offline dataset. See Appendix D.1 and Algorithm 1 for details.
>
> > The notation in Figure 1 is not very clear to me. Is it suppose to visualize Eq. 4?
>
> Thanks for the suggestion! Figure 1 is supposed to visualize a sample-based version of Eq. 4 and our theoretical analysis in Appendix A also indicates that TD InfoNCE builds upon a Bellman equation similar to the one in this figure. The color of the edges should correspond to different terms in Eq. 4. We have updated Eq. 4 and Fig. 1 to make the notations consistent.
>
> > In Eq. 7 and the one above, should it be $p^{\pi}(s_{t+}^{(1)} \mid s’, a’, g)$ instead of $p^{\pi}(s_{t+}^{(1)} | s’, a’)$?
>
> Thanks for finding the typo. We have updated the notation in Eq. 8.
>
> > In Table 1, the result for contrastive RL on large-diverse-v2 should not be bold; result for TDInfoNCE on unmaze-v2 should not be bold.
>
> Thanks for the suggestion. In Table 1, we bold the success rates within one standard deviation of the best methods. In this case, the result for contrastive RL on $\texttt{large-diverse-v2}$ should be bold and the result for TD InfoNCE on $\texttt{unmaze-v2}$ should not be bold. We have revised the table.

---

> > ### Comment · Reviewer_a2XQ · 2023-11-23
> > **Thanks for the reply**
> >
> > I would like to thank the author for the response. The updates did make the paper clearer. I would maintaining my score.

---

### Official Review · Reviewer_2Mg8 · 2023-10-31

**Soundness:** 3 good
**Presentation:** 3 good
**Contribution:** 3 good
**Rating:** 6
**Confidence:** 4

**Summary:**

This paper derives a TD variant of the InfoNCE objective function, relating this to some of the work in reinforcement learning using future distributions of state as an objective (i.e. successor representations/features). This algorithm is then applied to goal-conditioned reinforcement learning, showing competitive performance among several baselines.

**Strengths:**

The derived method fits nicely within the literature and seems to fill a nice gap between contrastive objectives from self-supervised objectives and more online focused temporal-difference updates.

**Weaknesses:**

After the rebuttal, I'm raising my score. While I believe there are issues with empirical section still, these are issues the rest of the literature are also facing. I don't think rejecting this paper is a way to a solution. I also appreciate the fix to some of the inaccurate statements that were overlooked!

Great job authors!

--------before edit-------

This paper struggles with clarity and accuracy in some of the ancillary statements made about the literature surrounding the paper and in the main content of the paper itself. There are also several issues with the experimental section that should be addressed.

## Accuracy and Clarity:
1. On page 1, in the paragraph starting with “the key aim…”, the language on motivating why a TD version of InfoNCE is oddly phrased. I think a fix should be easy here by removing phrases such as “may allow”, “may enable” and be more actionable in your language. This could also be replaced by actual hypotheses of what you expect to see from your objective and stated more formally.
2. In the same paragraph, and further throughout, you make a statement which suggests TD can do counter factual reasoning (or in the parlance of RL make use of off-policy updates) while Monte Carlo estimates cannot.  This is not true as Monte Carlo estimates can be made with off-policy corrections (using Importance Sampling like in TD). While this induces a more variant estimate as compared to TD (there is a classic bias-variance trade-off between MC and TD updates) and a TD update enables the use of incomplete trajectories (because you are using an estimate to inform your update rather than a full trajectory), I came out of the paper with the feeling the paper was suggesting Monte Carlo estimates couldn’t be off-policy.
    - This comes up very strongly on page 5 (the first paragraph) “This is, we cannot share…”. We should be able to derive an off-policy version of the monte-carlo update for InfoNCE. This doesn’t mean it would be an estimator we would want to use in this setting, but it should be definable. If this is not the case, the paper should show that this can’t be done using importance sampling or cite a reference which shows monte carlo estimates can’t be off-policy.

3. **Notation clarity issues:**
   - In your expectations above equation 7, I’m not sure what s’, a’ are here. Are you using these instead of s_{t+1} as used in equation 4? This notation should be unified.
   - Equation (1) and following uses, I’m not sure you explain what the superscript is signifying. I think it is time, but it is not clear from the writing.
   - Shouldn’t the expectation in the middle of page 4 on the RHS (i.e. after applying the importance weight) be selecting actions a’ from the behavior policy? Or is the importance weight only correcting for the state distribution? Shouldn’t we also correct for the action distribution as well?



## Empirical Results:

There are two major issues with the empirical results as presented.

### Major issues:

4. 3 seeds is too few to get any statistical confidence, especially without doing independent hyperparameter sweeps for each baseline. While in the past this has been standard, as a field we continually have shown that the statistical power of our experiments are laughably poor, even if the bounds of our results show statistical significance. This continually misleads the community, and needs to be addressed. This doesn’t include the issue with not running hyperparameter studies on the methods independently. See https://sites.ualberta.ca/~amw8/cookbook.pdf for a reference. While this paper is a draft it does a good job going through these issues and providing context from the literature.

5. In the appendix it is mentioned “We increase the capacity of the goal-conditioned state-action encoder…”. This suggests the model you are using may have more parameters than your counterparts. Is this true? Also did you use a larger batch size for all baselines or just your algorithm? If this was done for your method, this makes the results difficult to trust. If not, it likely means the hyperparameters of your baselines are now invalid. If both methods have the same hyperparameters then I’m usually ok with re-using old hyperparameter studies, unfortunately when any of the hypers change OR the new method has additional hypers this begins to weaken the validity of re-using the same hypers.

### Minor Issues:
- You should include all hyperparameters you used, even for baselines.
- You include only success rate as a metric to compare. While this is reasonable, I think there is a lot that could be learned from more traditional metrics (i.e. return or something similar). This is especially the case when the success metric doesn’t clearly separate the methods (for instance reach, Push, slide (state) in firgure 2a).



# Edits/Suggestions
- I don’t like the notation $s_{t+}$. I think it could be replaced with something that is more understandable on first read (and looks less like a mistake). This is a preference though.
- Page 4: “Our proposed method (Sec 3)…” Should say Sec 3.2.
- Page 4: “we conjecture that our…” -> You should state that you test this in the empirical section (I think you do at least).
- Page 7: “We will to evaluate” -> “we will evaluate”
- If you reference a result in the main paper you should include this in the main paper. “TD InfoNCE achieves a 2x median improvement etc…” references figure 6 (I believe).

**Questions:**

See above.

---

> ### Author Response · Authors · 2023-11-19
> **Rebuttal by Authors - Part 1**
>
> We thank the reviewer for the detailed response and for the suggestions for improving the paper. To address the concerns about the experiments, we have run additional random seeds and clarified that the baselines mostly use the same number of parameters and batch size as our method. We have also revised the paper to address concerns about clarity. **Together with the discussion for other questions below, does this fully address the reviewer’s concerns about the paper?**
>
> > This doesn’t include the issue with not running hyperparameter studies on the methods independently.
>
> As suggested by the reviewer, we ran additional experiments to study the effect of different hyperparameters for TD InfoNCE in table 2. For each hyperparameter, we selected a set of different values and conducted experiments on $\texttt{push (state)}$ and $\texttt{slide (state)}$, one easy task and one challenging task, for five random seeds. We report mean and standard deviation of success rates in Appendix E.1. These results suggest that while some values of the hyperparameter have similar effect, e.g. actor learning rate = $5 \times 10^{-5}$ vs $1 \times 10^{-4}$, our choice of combination is optimal for TD InfoNCE.
>
> > This suggests the model you are using may have more parameters than your counterparts. Is this true?
>
> For fair comparisons, we also increased the size of neural networks in baselines to the same number whenever possible in the original submission. We have added a sentence in Appendix D.5 to clarify. We list the number of parameters for each method evaluated on online GCRL benchmarks in the table below.
>
> | | TD InfoNCE | QRL | Contrastive RL | Contrastive RL (NCE) | GCBC | DDPG + HER|
> |:---------------------------:|:---------------------:|:-----------------:|:---------------------:|:---------------------:|:---------------------:|:---------------------:|
> |   number of parameters    |    $1,629,184$   | $2,147,328$ | $1,629,184$ | $1,629,184$ | $1607680$ | $1,629,022$ |
>
> As shown in the table, TD InfoNCE mostly uses the number of parameters less than or equal to baselines. Note that QRL contains a feedforward encoder $f$, a residual latent transition $T$, and a projector $d_{\theta}$ in its critic (Appendix C.3 of [2]). Even if we set the size of $f$ and $T$ to be the same as the networks in TD InfoNCE, the number of parameters in QRL is still larger than that of TD InfoNCE.
>
> > Also did you use a larger batch size for all baselines or just your algorithm?
>
> All baselines used the same batch size as our method: 256. We have added this detail to Appendix D.5.
>
> > 3 seeds is too few to get any statistical confidence …
>
> As suggested, we have repeated experiments in Fig. 2 (Appendix Fig. 6) and Table 1 with two more random seeds (five in total) to increase the statistical confidence of our experiments. We report results on the Fetch robotics benchmark and D4RL benchmark in the revised paper. These results suggest that TD InfoNCE is stable across different random seeds and our conclusions in Sec. 4.1 and Sec. 4.2 remain the same.
>
> > This is not true as Monte Carlo estimates can be made with off-policy corrections (using Importance Sampling like in TD).
>
> Thanks for catching this. We agree that Monte Carlo estimates can be used in an off-policy manner by applying importance weights to correct actions. We have revised the text in the final paragraph of Sec. 3.1 to note that MC methods with importance weighting could in theory perform off-policy evaluation (specifically, off-policy contrastive RL in our problem), while citing prior work that shows that off-policy evaluation based on importance weights tends to be unstable [5, 6].
>
> > … the language on motivating why a TD version of InfoNCE is oddly phrased.
>
> We have revised this sentence in the introduction.
>
> > In your expectations above equation 7, I’m not sure what s’, a’ are here. Are you using these instead of s_{t+1} as used in equation 4? This notation should be unified.
>
> $s’$ is the next state sampled from the environment while $a’$ is the corresponding action sampled from the goal-conditioned policy given $s’$. We have revised equation 4 to make the notation consistent.
>
> > Equation (1) and following uses, I’m not sure you explain what the superscript is signifying. I think it is time, but it is not clear from the writing.
>
> In equation 1, we sample one "positive" $y$ (denoted $y^{(1)}$) and $N - 1$ "negative" $y$s (denoted $y^{(2:N)} = \\{y^{(2)}, \cdots, y^{(N)} \\}$). We follow the notational convention of prior work [1, 3, 4]. We have clarified this detail in the sentence before equation 1.

---

> > ### Author Response · Authors · 2023-11-19
> > **Rebuttal by Authors - Part 2**
> >
> > > Shouldn’t the expectation in the middle of page 4 on the RHS (i.e. after applying the importance weight) be selecting actions a’ from the behavior policy? Or is the importance weight only correcting for the state distribution? Shouldn’t we also correct for the action distribution as well?
> >
> > We assume that the reviewer was referring to the equation of $L_2(f)$ (equation above Eq. 8 in page 5); please let us know if we are mistaken. If we sample the next action a’ from the behavior policy $\beta(a \mid s)$, the algorithm will be similar to SARSA, estimating the discounted state occupancy measure of the $\textit{behavioral policy}$. Since our aim is to estimate the discounted state occupancy measure of the $\textit{target policy}$, we use the target policy in our TD target, resulting in an algorithm similar to Q-Learning.
> >
> > The expectation over next state $s’ \sim p(s’ \mid s, a)$ and next action $a’ \sim \pi(a’ \mid s)$ in $L_2(f)$ comes from the second term of Eq. 4 and it only accounts for sampling future states from the discounted state occupancy measure of the target policy starting from the next time step $s_{t+}^{(1)} \sim p^{\pi}(s_{t+} \mid s’, a’) $. However, this procedure still requires sampling from the quantity we are trying to estimate and we use importance sampling weight $p^{\pi}(s_{t+}^{(1)} \mid s’, a’) / p(s_{t+}^{(1)})$ to avoid it. Since we only consider one transition, we don’t need an importance weight for actions. Note that there is another level of expectation outside $L_2(f)$ sampling $(s, a)$ and $s_{t+}^{(2:N)}$.
> >
> > > You should include all hyperparameters you used, even for baselines.
> >
> > As mentioned in Appendix D.5, we strictly follow the baseline implementations provided by the author of contrastive RL [1] and QRL [2]. Hyperparameters of those baselines can be found in Appendix E of [1] and Appendix C.3 of [2]. Code for our method and baselines are included in the footnotes of Sec. 3.3 and Appendix D.5.
> >
> > > You include only success rate as a metric to compare. While this is reasonable, I think there is a lot that could be learned from more traditional metrics (i.e. return or something similar). This is especially the case when the success metric doesn’t clearly separate the methods (for instance reach, Push, slide (state) in figure 2a).
> >
> > As suggested by the reviewer, we compare TD InfoNCE to QRL and contrastive RL using minimum distances of the gripper or the object to the goal over an episode as the metric in the new Appendix Fig. 7. Note that a lower minimum distance indicates a better performance. Using the minimum distance, we found that TD InfoNCE did perform better than those two baselines on tasks where success rates were not separable (e.g., $\texttt{slide (state)}$ and $\texttt{push (image)}$). These results suggest that TD InfoNCE is able to emerge a goal-conditioned policy by estimating the discounted state occupancy measure, serving as a competitive goal-conditioned RL algorithm. We have revised Sec. 4.1 and Appendix D.2.
> >
> > We also note that the return is equivalent to the success rate in goal-conditioned RL problems (See the second paragraph of Sec. 3.1 for a discussion).
> >
> > > Page 4: “Our proposed method (Sec 3)…” Should say Sec 3.2. Page 7: “We will to evaluate” -> “we will evaluate”
> >
> > Thanks for the suggestions! We have revised the paper.
> >
> > > Page 4: “we conjecture that our…” -> You should state that you test this in the empirical section (I think you do at least).
> >
> > We have added a reference in this sentence to experiments in Appendix E.3.
> >
> > > If you reference a result in the main paper you should include this in the main paper. “TD InfoNCE achieves a 2x median improvement etc…” references figure 6 (I believe).
> >
> > We have revised this sentence to indicate that the result can be seen in Appendix Fig. 6..
> >
> > [1] Eysenbach, B., Zhang, T., Levine, S. and Salakhutdinov, R.R., 2022. Contrastive learning as goal-conditioned reinforcement learning. Advances in Neural Information Processing Systems, 35, pp.35603-35620.
> >
> > [2] Wang, T., Torralba, A., Isola, P. and Zhang, A., 2023. Optimal Goal-Reaching Reinforcement Learning via Quasimetric Learning. arXiv preprint arXiv:2304.01203.
> >
> > [3] Oord, A.V.D., Li, Y. and Vinyals, O., 2018. Representation learning with contrastive predictive coding. arXiv preprint arXiv:1807.03748.
> >
> > [4] Poole, B., Ozair, S., Van Den Oord, A., Alemi, A. and Tucker, G., 2019, May. On variational bounds of mutual information. In International Conference on Machine Learning (pp. 5171-5180). PMLR.
> >
> > [5] Precup, D., Sutton, R.S. and Singh, S.P., 2000, June. Eligibility Traces for Off-Policy Policy Evaluation. In Proceedings of the Seventeenth International Conference on Machine Learning (pp. 759-766).
> >
> > [6] Precup, D., Sutton, R.S. and Dasgupta, S., 2001, June. Off-Policy Temporal Difference Learning with Function Approximation. In Proceedings of the Eighteenth International Conference on Machine Learning (pp. 417-424).

---

> > > ### Author Response · Authors · 2023-11-21
> > > **Do the new experiments and revisions address the concerns?**
> > >
> > > Dear Reviewer,
> > >
> > > **Do the new experiments and revisions described above fully address the concerns about the paper?** We believe that these changes, motivated by the excellent feedback, further strengthens the paper. While the time remaining in the review period is limited, we would be happy to try to run additional experiments or make additional revisions.
> > >
> > > Kind regards,
> > >
> > > The Authors

---

> ### Comment · Reviewer_2Mg8 · 2023-11-21
> **Response**
>
> Thank you for the detailed response! I think you have done a great job addressing my concerns! I think you have clarified my questions and concerns. While the empirical section is still not up to the standards I would hope, this is more inline with the literature now.
>
> Great job!

---

### Official Review · Reviewer_yDjK · 2023-11-09

**Soundness:** 3 good
**Presentation:** 3 good
**Contribution:** 3 good
**Rating:** 8
**Confidence:** 3

**Summary:**

Predicting future states is crucial for many time-series tasks, including goal-conditioned reinforcement learning (RL). While contrastive predictive coding (CPC) has been used to model time series data, learning representations that capture long-term dependencies often requires large datasets. This paper introduces a temporal difference (TD) version of CPC that combines segments of different time series, reducing the data needed to predict future events. This representation learning method is applied to derive an off-policy algorithm for goal-conditioned RL. Experiments show that the proposed method achieves higher success rates with less data and better handles stochastic environments compared to previous RL methods.

**Strengths:**

- The paper proposes a new temporal difference (TD) estimator for the InfoNCE loss, which is shown to be more efficient than the standard (Monte Carlo) estimator.

- The proposed goal-conditioned reinforcement learning (RL) algorithm outperforms prior methods in both online and offline settings.

- The proposed algorithm is capable of handling stochasticity in the environment dynamics.

- In stochastic tasks, there is an excellent improvement in performance versus the baseline of Quasimetric RL, with some healthy gains on non stochastic tasks versus other baselines, although this is not the primary target of the paper.

- The paper provides a clear and concise explanation of the proposed algorithm.

- The paper is well-written and easy to understand.

- The paper is well-supported by experiments.

- The proposed algorithm is evaluated on a variety of tasks, including the Fetch robotics benchmark.

- TD InfoNCE learns on image-based pick & place and slide, while baselines fail to make any progress.

- TD InfoNCE maintains high success rates on more challenging tasks with observation corruption, while the performance of QRL decreases significantly.

**Weaknesses:**

- The paper focuses on fairly trivial environments, it would be nice to see these methods working on more challenging higher dimensional goal conditioned RL tasks, as its not a given that these gains will carry over to tasks that matter a lot more.

- The proposed TD estimator is more complex than the standard (Monte Carlo) estimator and its implementation requires more hyperparameters.

- The performance of the proposed goal-conditioned RL algorithm on the most challenging tasks was less than 50%.

- QRL assumes deterministic dynamic of the environment, while TD InfoNCE learns without such assumption.

Loss Function Composition: The loss function L(θ) is composed of two cross-entropy (CE) loss terms, one for predicting the next state and one for predicting the future distribution of states. The γ hyperparameter is used to weight these two terms, but the choice of γ and its impact on the algorithm's performance are not discussed in detail.

**Questions:**

Can you explain how you selected the hyperparameters for the proposed algorithm?

Can you provide more details about the observation that TD InfoNCE learns on image-based pick & place and slide, while baselines fail to make any progress?

---

> ### Author Response · Authors · 2023-11-19
> **Rebuttal by Authors**
>
> We thank the reviewer for the responses and suggestions for improving the paper. The reviewer brought up two questions regarding the experiments: selection of hyperparameters and explanation of an observation on challenging image-based tasks. We have attempted to address the question about hyperparameter selection by running ablation experiments, and we answer the question about the performance of TD InfoNCE below. **Together with the discussion below, does this fully address the reviewer’s questions?**
>
> > Can you explain how you selected the hyperparameters for the proposed algorithm?
>
> We ran ablation experiments to find the best hyperparameters for TD InfoNCE (new Appendix E.1). Since our algorithm shares the same set of hyperparameters with contrastive RL, we started with the default hyperparameters from this prior work. Among those hyperparameters we chose to sweep over actor learning rate, critic learning rate, representation normalization, MLP hidden layer sizes, and representation dimensions in a range of different values (see new Appendix Fig. 10). These results suggest that while some values of the hyperparameter have similar effect, e.g. actor learning rate = $5 \times 10^{-5}$ vs $1 \times 10^{-4}$, our choice of combination is optimal for TD InfoNCE.
>
> > The γ hyperparameter is used to weight these two terms, but the choice of γ and its impact on the algorithm's performance are not discussed in detail.
>
> The $\gamma \in (0, 1]$ is the discount factor of the underlying Markov decision process. We ran ablation experiments with $\gamma \in \\{0.90, 0.95, 0.99\\}$. Results in Appendix Fig. 10 suggests that a larger discount factor achieves a higher success rate, which can be explained by focusing more on (higher weight) correctly predicting which states are likely future states (Eq. 10).
>
> > Can you provide more details about the observation that TD InfoNCE learns on image-based pick & place and slide, while baselines fail to make any progress?
>
> The image-based $\texttt{pick \\& place}$ and $\texttt{slide}$ tasks are the most challenging tasks we consider – not only do they involve manipulating other objects, but also require learning such behavior from high-dimensional sensory input (i.e., images). We believe that TD InfoNCE outperforms prior methods on this task because it is able to make better use of limited data (e.g., through the temporal difference updates).
>
> > The paper focuses on fairly trivial environments, it would be nice to see these methods working on more challenging higher dimensional goal conditioned RL tasks …
>
> Our current set of experiments on 14 tasks (8 on the Fetch robotics benchmark, 6 on the D4RL benchmark) includes tasks with 12288-dimensional (64 x 64 x 3) image observations. Prior methods struggle on at least some of these tasks (\texttt{pick & place (image)} and \texttt{slide (image)}), suggesting that these tasks are at least challenging for prior methods. While this breadth of tasks at least matches that of prior work [1, 2], we would be happy to investigate additional challenging tasks if suggested by the reviewer.
>
> > QRL assumes deterministic dynamics of the environment, while TD InfoNCE learns without such assumption.
>
> We believe this is a strength of TD InfoNCE, rather than a weakness. In general, making assumptions about the dynamic of the environment narrows the domain where an RL algorithm might be applicable. Fig. 2b shows that TD InfoNCE can solve some tasks with stochastic dynamics while QRL suffers from a significant drop in performance compared to its result on the deterministic version. We suspect that TD InfoNCE can be applied to a wider class of goal-conditioned RL problems.
>
> [1] Eysenbach, B., Zhang, T., Levine, S. and Salakhutdinov, R.R., 2022. Contrastive learning as goal-conditioned reinforcement learning. Advances in Neural Information Processing Systems, 35, pp.35603-35620.
>
> [2] Wang, T., Torralba, A., Isola, P. and Zhang, A., 2023. Optimal Goal-Reaching Reinforcement Learning via Quasimetric Learning. arXiv preprint arXiv:2304.01203.

---

### Author Response · Authors · 2023-11-19
**List of Changes in the Rebuttal Version**

We thank the reviewers for their feedback that helps to improve the paper. We have revised the paper to add some new results and incorporate suggestions from the reviewers (orange texts). Below, we summarize the changes of the rebuttal version.
- We have revised the notations in Sec. 3 to remove the dependence of the critic on goals and the goal-conditioned policy: $f(s, a, g, s_{t+}) \to f(s, a, s_{t+})$ and $\pi(a \mid s, g) \to \pi(a \mid s)$. There are three main reasons for these revisions. First, these revisions emphasize the motivation of building a temporal difference estimate of the discount state occupancy measure for $\textit{any}$ policy, not necessary depending on a goal. Second, deriving a goal-conditioned RL algorithm is an application of TD InfoNCE which can be achieved by substituting state $s$ with a pair of state and goal $(s, g)$ and replacing $\pi(a \mid s)$ with $\pi(a \mid s, g)$ in our notations. Finally, our notations imply that TD InfoNCE can potentially be applicable to other problems, e.g., time series analysis, structural representation learning on CV and NLP datasets, and off-policy policy evaluation.
- We have moved MC InfoNCE from the discussion of the main methods (Sec. 3.2) into the last paragraph of the preliminary since MC InfoNCE has already been introduced in prior work: contrastive RL (CPC) in [1].
- In Sec. 3.2 and Appendix A, we have added a proof of convergence of (a variant of) our method in tabular settings, showing that TD InfoNCE and MC InfoNCE converge to the same optimal critic. This theoretical analysis strengthens the soundness of our estimator.
- In addition, the theoretical analysis in Appendix A also helps simplify the derivation of the connection between TD InfoNCE and successor representations (see Appendix C), showing that successor representation update is a special case of TD InfoNCE in tabular settings.
- Also, the connection of TD InfoNCE and mutual information maximization (see Appendix B) becomes revealing since prior work has proved that MC InfoNCE is a lower bound of mutual information between input pairs [2].
- We include a new representation interpolation experiment in Appendix E.4 showing the representations learned by TD InfoNCE captures causal relationships.
- We add hyperparameter ablation experiments in Appendix E.1.
- We add a comparison to MC InfoNCE (or contrastive RL (CPC)) [1] in Fig. 2 and Appendix Fig. 6. Note that we rename the original contrastive RL baseline to contrastive RL (NCE).
- For those tasks where success rates are not separable for different methods, we add comparisons using minimum distance of the gripper or the object to the goal over an episode in Appendix D.2.
- As suggested by the reviewers, we have incorporated their feedback into the corresponding texts in the paper (orange texts). See response below for details.

[1] Eysenbach, B., Zhang, T., Levine, S. and Salakhutdinov, R.R., 2022. Contrastive learning as goal-conditioned reinforcement learning. Advances in Neural Information Processing Systems, 35, pp.35603-35620.

[2] Poole, B., Ozair, S., Van Den Oord, A., Alemi, A. and Tucker, G., 2019, May. On variational bounds of mutual information. In International Conference on Machine Learning (pp. 5171-5180). PMLR.

---

### Meta-Review · Area_Chair_dXvD · 2023-12-11

**Metareview:**

This paper introduces a TD version of contrastive predictive coding, in particular InfoNCE, aiming to improve performance and sample efficiency. Their method shows good results over a wide variety of baseline approaches.

The reviewers appreciated the overall clarity of the writing, the presentation of a theoretical foundation and derivation for the method, comprehensive nature of their experiments, and the strength of the results. The main issues that were raised centered around 1) missing experimental details such as hyperparameters and hyperparameter selection, 2) only running on 3 seeds, and 3) some missing comparisons and ablations.

The authors were highly responsive and added several new experiments as well as the missing experimental details and additional seeds, which for the most part satisfied reviewers and led to a much stronger paper. In the end a consensus was reached for acceptance.

**Justification For Why Not Higher Score:**

The approach is in the end only an extension of InfoNCE, and is evaluated on a small set of benchmark RL tasks, so it's unclear if it's widely applicable.

**Justification For Why Not Lower Score:**

This is an overall pretty solid paper. The approach is theoretically grounded and backed up with comprehensive experiments, and authors satisfied almost all reviewer concerns.

---

### Decision · Program_Chairs · 2024-01-16

Accept (poster)